# LinRace: cell division history reconstruction of single cells using paired lineage barcode and gene expression data

Xinhai Pan [1], Hechen Li [1], Pranav Putta[1] & Xiuwei Zhang [1] ✉

Lineage tracing technology using CRISPR/Cas9 genome editing has enabled simultaneous readouts of gene expressions and lineage barcodes in single cells, which allows for inference of cell lineage and cell types at the whole organism level. While most state-of-the-art methods for lineage reconstruction utilize only the lineage barcode data, methods that incorporate gene expressions are emerging. Effectively incorporating the gene expression data requires a reasonable model of how gene expression data changes along generations of divisions. Here, we present LinRace (Lineage Reconstruction with asymmetric cell division model), which integrates lineage barcode and gene expression data using asymmetric cell division model and infers cell lineages and ancestral cell states using Neighbor-Joining and maximum-likelihood heuristics. On both simulated and real data, LinRace outputs more accurate cell division trees than existing methods. With inferred ancestral states, LinRace can also show how a progenitor cell generates a large population of cells with various functionalities.

Understanding how cells divide and differentiate into various cell types is a fundamental problem in developmental biology. *Lineage tracing* technology which traces cell divisions using a "recorder" is the most widely used technique to study the developmental histories of cells, while traditional lineage tracing technologies can only work with a limited number of cells with low resolution[1]. Recently, sequencing-based lineage tracing methods (e.g., using CRISPR/Cas9 genome editing) have enabled the simultaneous recording of the clonal relationships of single cells alongside the transcriptomes[2] for up to thousands of cells. Such methods utilize lineage recorders, which are exogenous DNA sequences integrated into the genome. Even though there are different ways of designing the lineage recorder[3-11], the common idea is to introduce changes at the target sites (the location to which the Cas9 protein binds to induce mutations) on the lineage recorder, which accumulates through generations of cell divisions. Finally, the recorders are sequenced together with the transcriptome of every single cell, resulting in the "lineage barcode" data. Each target site corresponds to a character in the barcode. The barcode data are used to reconstruct the cell division tree, which is also called the cell lineage tree in this paper. In this paper, we use "cell lineage tree" and

"cell division tree" interchangeably, both of which denote the directed, binary tree graph of cells' division history. Each node represents a single cell and each edge represents a direct parent-daughter cell relationship in the cell division event. The reconstructed cell division tree can shed light on the developmental process that can not be directly measured.

However, inferring the cell division tree of a massive number of cells is a challenging problem. In addition to the computational complexity of inferring the lineage tree itself, the quality of the barcode data has posed a further challenge to this problem[12]. First, the number of target sites, which is the length of the string used for tree reconstruction, is usually small (the length of the string is 18 in a mouse embryo dataset[5] and 9 in a zebrafish dataset[9]); Second, dropouts in the CRISPR/Cas9 induced lineage barcode data can cause missing information in the data. There are two types of dropouts: one is called *excision dropout*, or collapse dropout[13], result in the loss of consecutive target sites in between two simultaneous mutations (in this case, the mutations are deletions in the barcode). The other type of dropout is due to the limited capture efficiency of the sequencing experiment, where the barcodes of certain cells are not profiled. Finally, the biased

---

[1]School of Computational Science and Engineering, Georgia Institute of Technology, Atlanta, GA 30332, USA. ✉e-mail: xzhang954@gatech.edu

distribution of mutations across the barcode and the number of mutations, represented by the *mutation rate* parameter, may not be optimal for reconstructing the cell division tree. It is shown that the distribution of mutations across the barcode is not uniform, but rather biased towards certain target sites[14,15]. The mutation rate, which represents the efficiency of mutations being induced in the barcode, has a major impact on the potential to successfully infer the cell division lineages from the data. However, current experimental technologies do not guarantee that the mutations occur at a rate that allows the tracing of every cell division event.

Various methods for tree inference from lineage barcodes have been developed. Recently, a DREAM challenge was held to gather the community effort to compare the state-of-the-art lineage tree inference methods[16]. Among the benchmarked algorithms, the best performers are: DCLEAR[17], a distance-based method that first calculates the pairwise distance between cells and then reconstructs the cell lineage using bottom-up (agglomerative) algorithms such as Neighbor Joining (NJ)[18] or FastME[19]; Cassiopeia[20], a parsimony-based method that aims at minimizing the number of mutations occurred on the reconstructed lineage tree. A recent method, Startle[21], aims to infer cell division trees from lineage barcode data by enforcing the "non-modifiability" property of CRISPR/Cas9 mutations. However, current methods for cell lineage tree reconstruction do not provide satisfying results using barcode data[14–16]. Moreover, due to the short barcode length and dropouts, a number of cells can have the same barcode, and the reconstructed lineage trees tend to have low depth (maximum path length from the root to a leaf node) and few internal nodes (with some nodes having large degrees) even using a perfect method. More recently, methods that combine lineage barcode and gene expression data are proposed, aiming to further improve the accuracy of cell lineage reconstruction. LinTIMaT[22] develops a combined likelihood function and uses a local search framework to search for the tree with the maximum likelihood. Integrating paired gene expression to infer the cell lineage tree can potentially refine the reconstruction, however, despite that the paired data should theoretically provide more information than barcodes alone, LinTIMaT did not beat methods that use only the lineage barcode data according to previous comparisons on synthetic datasets[14].

The key to combining the lineage barcode and gene expression data is to model the relationship between the two types of data, i.e., how the gene expression of cells changes along with the barcode data during cell divisions. However, although methods that combine lineage barcode and gene expression data have been proposed for other purposes (instead of the inference of cell lineage trees)[23,24], it is still an open question how the cell's transcriptome changes during cell division. A simple assumption is to assume that cells that have similar transcriptomes should be located close in the cell division tree, i.e., they should have similar barcodes. This is the assumption used in LinTIMaT. However, in a few recent papers that present paired single-cell lineage barcode and gene expression data, it is observed that, in a tree reconstructed using the barcode data, although a proportion of cells with the same cell state located in the same subtree, it is remarkable that some cells of the same cell state are located in different subtrees, and the same subtree can have multiple cell types[5,9]. We call this phenomenon the "partial consistency between transcriptome similarity and barcode similarity".

The *asymmetric cell division model* has been shown to be able to account for the "partial consistency between transcriptome similarity and barcode similarity"[14]. It is commonly considered that cells can divide in a symmetric or asymmetric manner[25,26]. A symmetric cell division gives rise to daughter cells with the same cell state as the parent cells. During an asymmetric cell division, one daughter cell keeps the parent's cell state, and the other one differentiates into a future cell state according to the *cell state tree*. In a *cell state tree*, each node represents a cell state and each edge represents a direct state transition from one cell state to another. For a given cell, there is a probability with which it divides asymmetrically, termed as the *asymmetric division rate*, denoted by $p_a$. It has been shown that this probabilistic asymmetric cell division model leads to realistic paired single-cell lineage barcode and gene expression data.

To address these problems, we present LinRace (Lineage Reconstruction with asymmetric cell division model), a method that combines the lineage barcode and gene expression data to infer cell division trees, based on a joint Neighbor Joining (NJ) and maximum-likelihood framework. The asymmetric cell division model is used in LinRace to infer the states of ancestral cells and to calculate the likelihood function, thus incorporating the relationship between lineage barcode and gene expression data realistically. On both simulated and real datasets, LinRace consistently outperforms the state-of-the-art methods according to multiple metrics. We show that the use of gene expression data in LinRace helps to improve the lineage tree reconstruction accuracy compared to methods that use only the lineage barcode data (Cassiopeia and DCLEAR). We also show that LinRace achieves better performances than the existing method that also uses gene expression data (LinTIMaT) while improving computational efficiency. Moreover, we demonstrate that when applied to large-scale real datasets, LinRace uncovers more detailed local lineage structures compared to LinTIMaT, as well as ancestral state information and state-lineage relationships that are consistent with observations from real data.

It is worth noting that the task of reconstructing the cell division trees is sometimes referred to as "lineage reconstruction". This is not to be confused with the computational task "trajectory inference", which infers continuous cell state changes among the cells in the single-cell gene expression data and often outputs a graph representing the changes between cell states (e.g., the cell state tree we use in this paper) and pseudotime of every single cell.

## Results

### Overview of LinRace

LinRace is motivated by the fact that currently available lineage barcode data cannot label each cell with a unique barcode, and can contain large subgroups of cells with the same barcode. For example, the embryo2 dataset in the mouse lineage tracing system[5] has 19,019 cells and 2788 unique barcodes. That means, a lot of cells are not uniquely labeled by one barcode. Indeed, 86.4% of the cells share the same barcode with at least two other cells (Supplementary Fig. 1a, top plot). In detail, when sorting the barcodes by the number of cells each barcode corresponds to, there are 529 barcodes, each of which corresponds to more than 3 cells, and these 529 barcodes together label 16,432 cells, with some barcodes labeling more than 1000 cells, resulting in large clones (one clone is defined as the cells sharing the same barcode). A detailed view of the top 50 barcodes with the largest clone sizes is shown in Supplementary Fig. 1a, bottom plot. The barcode data alone does not allow the inference of trees for the cells with the same barcode, moreover, the barcode homoplasy effect can lead to false grouping of cells from different clones. LinRace aims to infer the cell lineage tree using paired lineage barcodes and gene expression data, where every single cell has its corresponding lineage barcode and gene expression data. The overview of LinRace is shown in Fig. 1.

First, from the barcode data of all the cells, we extract the unique barcodes and build a tree of these barcodes (Fig. 1, Step A) using the neighbor-joining[18] tree reconstruction method. We then obtain a tree where each leaf node represents a unique barcode and can correspond to multiple cells with the same barcode. We denote this tree by $\mathcal{T}_0$. Then we use the single-cell gene expression data to refine the tree $\mathcal{T}_0$. We consider that there exists an underlying cell state transition mechanism that can be represented by a *cell state tree*, following which different cell types emerge during the cell division processes. This cell state tree can be inferred from the single-cell gene expression data

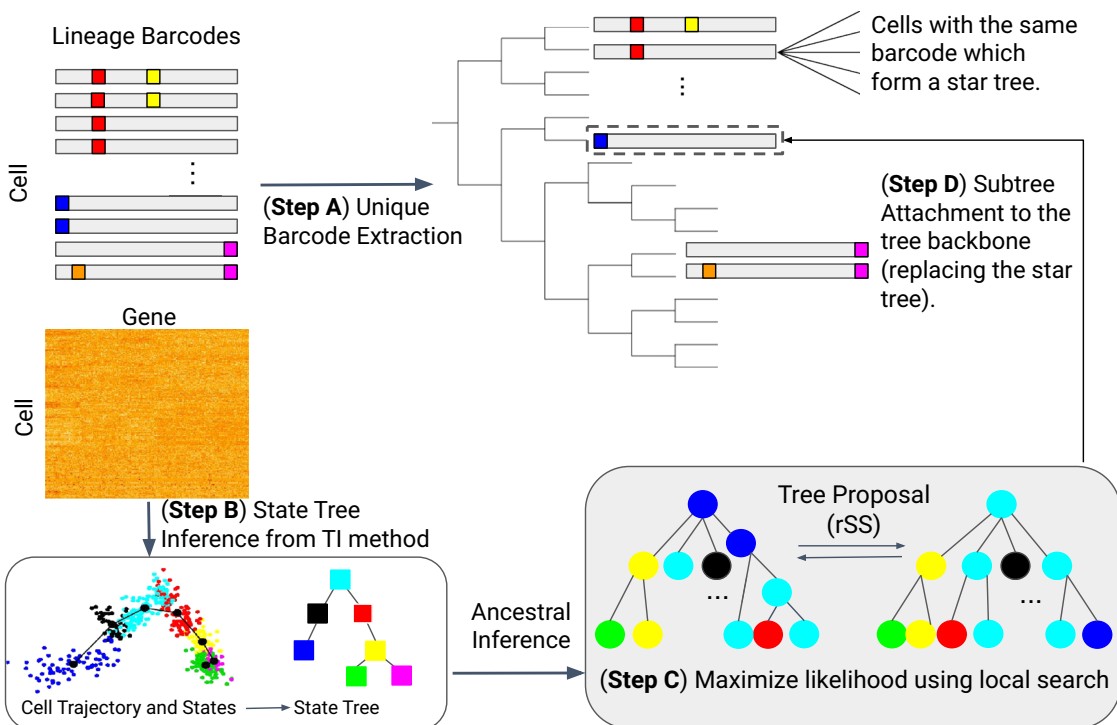

**Fig. 1 | Overview of LinRace.** Step A: For barcode data, we extract the unique barcodes and then perform Neighbor Joining to obtain the tree backbone, where each leaf represents a unique barcode shared by some cells. Step B: For the gene expression data, we use K-means on PCA reduced dimensions to infer the cell states, and then use Slingshot[28] to infer the state trajectories which are then used to infer the ancestral states. Step C: For each group of cells of the same barcode, we use a maximum likelihood + local search framework to find the subtree topologies of the cells. Step D: The final output tree is obtained by combining the subtrees at their specific leaves on the tree backbone. Nodes in the trees that are illustrated by squares represent cell states (or cell types), and those illustrated using circles represent cells, and the color of each circle node represents the state of the cell.

using trajectory inference methods[27–29] (Fig. 1, Step B). Assuming the cell state tree covers all cell states in the developmental process, the cell state tree along with the *asymmetric cell division* model together model the relationship between lineage barcode and the gene expression data and allow for the inference of cell states for ancestral cells, as well as the design of the likelihood function used in Step C of Fig. 1. Details on the motivation for using the asymmetric division model are in "Methods". Then, for each leaf in $\mathcal{T}_0$ which corresponds to one unique barcode and potentially many cells, we use a likelihood function ("Methods") to find the best binary tree (which means every non-leaf node has exactly two children nodes) that maximizes this likelihood (Fig. 1, Step C). This likelihood is based on the asymmetric cell division model and the cell state tree. Finally, we attach the subtrees inferred in Step C to the lineage tree inferred from the barcode data ($\mathcal{T}_0$) to get the complete lineage of single cells (Fig. 1, Step D).

### LinRace outperforms existing methods on synthetic datasets under various settings

We first test LinRace's tree reconstruction performance against baseline methods using simulated data. We use TedSim[14] to generate simulated paired single-cell gene expression and lineage barcode data, which is the only existing simulator that generates such paired data with a ground truth cell division tree. We compare the results of LinRace with the state-of-the-art lineage tree reconstruction methods, including Cassiopeia-greedy, Cassiopeia-hybrid[20], and Startle[21] (parsimony-based methods), and DCLEAR-kmer[17] (a distance-based method), which use only the barcode data, and LinTIMaT[22], a method based on the combined likelihood of gene expression and lineage barcode that use both types of data.

To obtain a comprehensive picture of the performances of the methods, we vary major parameters when generating the simulated data: (1) Mutation rate $\mu$, the probability that a mutation (insertion or

deletion) is induced per target site per cell division. Different mutation rates can result in barcode data with different quality, and it has been shown that the performance of lineage tree reconstruction methods using only the barcode data is affected by the mutation rate[14,15]. A range of [0.01, 0.3] was used in previous work[15], and we used a similar range, which is [0.03, 0.3]. (2) With or without dropout. Real data contains a significant amount of dropouts, and simulated data with dropouts show properties (e.g., the distribution of the number of cells with the same barcode in Supplementary Fig. 1b) much closer to real data than simulated data without dropouts. We nevertheless test the methods on simulated data without dropouts, which can show the effect of dropouts on each method. (3) Number of cells. The complexity of the lineage tree reconstruction problem increases with the number of cells. We generated datasets with 1024 and 4096 cells. For each combination of parameters, we generated ten datasets with ten random seeds. More parameter settings on data simulation and the simulation process are in "Methods".

The software versions and parameters used for the baseline algorithms are in "Methods". For LinRace, the major hyperparameters are $\lambda_1$ and $\lambda_2$, which are weights for different terms in the likelihood function ("Methods"). we use the default settings, $\lambda_1 = 10$ and $\lambda_2 = 1$, for all datasets. Although we provide default values for $\lambda_1$ and $\lambda_2$, we show that LinRace is not sensitive to changes in these parameters. In Supplementary Fig. 2, we show the performances of LinRace are similar under various parameter settings for $\lambda_1$ and $\lambda_2$. Other parameters used for LinRace are specified in "Methods".

As the states of cells and the cell state tree inferred in Fig. 1, Step B are used to infer the states of ancestral cells and calculate the likelihood (Fig. 1, Step C), the accuracy of the inferred cell states and cell state tree can affect the lineage tree reconstruction accuracy of LinRace. With the ground truth cell states and cell state tree provided by TedSim, we can investigate the effect of cell state and cell state tree

inference on the final performance of LinRace. To do this, we run LinRace in two modes: (1) using the ground truth cell states and the cell state tree, this mode is denoted as LinRace-TST (True State Tree); (2) using inferred cell states and the cell state tree, and this mode is denoted as LinRace-IST (Inferred State Tree). We use Slingshot[28] to infer cell states and the cell state tree from the gene expression data. As Slingshot does not infer the direction of trajectories, it is a common practice to assign a root cell so that we obtain a directed trajectory. We randomly selected a cell from the root cell state in the true cell state tree to provide this cell to Slingshot as the root cell. When working with real data, the root cell state needs to be given to Slingshot based on prior knowledge.

Evaluating the predicted cell division trees involves calculating the distance or similarity between the predicted and ground truth trees, which is not a trivial task due to the large variety of tree structures. Aiming to provide a comprehensive evaluation, we use three different metrics, RF (Robinson–Foulds) distance[30], Nye Similarity[31], and Clustering Info Distance (CID)[32] ("Methods"). For RF distance and CID, lower is better and for Nye Similarity, higher is better. The settings of data simulation and parameters for running all the methods are in Supplementary Note 2 and "Methods".

The results are shown in Fig. 2 (on datasets with dropouts) and Supplementary Fig. 3 (on datasets without dropouts). The results of LinRace-TST, LinRace-IST and baseline methods on all simulated datasets with 1024 cells are shown in Fig. 2a–c. Nye similarity and CID failed to run on the datasets with 4096 cells, so we present the RF distance for datasets with 4096 cells (Fig. 2d, Supplementary Fig. 3d). We also show the results on datasets with extended lengths of target sites (1024 cells and 128 target sites, with dropouts) in Supplementary Fig. 4. First, we observe that in Fig. 2a–c, the two modes of LinRace, LinRace-TST and LinRace-IST, clearly outperform all other methods, in terms of all three metrics on datasets with 1024 cells. On datasets with 4096 cells (Fig. 2d) and 128 target sites (Supplementary Fig. 4), LinRace-TST and LinRace-IST also outperform other methods using RF distance, with even more improvement compared to results on 1024 cells (Fig. 2a), confirming its effectiveness on large datasets. Second, most methods show a similar trend when the mutation rate changes, except for LinTIMaT. Comparing results on data with dropouts (Fig. 2) and without dropouts (Supplementary Fig. 3), most methods gain much better accuracy on data without dropouts, except for LinTIMaT. Overall, LinTIMaT is not sensitive to the quality of the barcode data, however, its performance is consistently worse than almost all other methods, including those using only barcode data, which indicates that it may not take full advantage of the barcode data. Furthermore, it uses a whole-tree local search strategy, which limits its effectiveness on large trees with thousands of leaves. This point can be further supported by the performances of Startle-NNI, which also uses whole-tree local search, with a NJ tree initialization. Compared to other barcode-based methods (DCLEAR and Cassiopeia, Fig. 2a–c), Startle is shown to be less accurate in most of the cases. One can potentially increase the number of NNI (Nearest-Neighbor-Interchange) iterations, but the program is already slow when setting it to 5000, which is far from enough to explore the tree search space. Finally, the tree reconstruction methods overall have better performance on barcode data with 128 target sites (Supplementary Fig. 4) compared to 16 target sites (Fig. 2a–d), suggesting that longer barcodes can help achieve more accurate trees, though experimentally obtaining barcode data with this length is still challenging.

By default, LinRace uses Neighbor Joining to build the backbone tree $\mathcal{T}_0$. In principle, other barcode-based lineage reconstruction methods can also be used to build the backbone tree. Here, we ran LinRace using different barcode-based methods (NJ, Cassiopeia, and DCLEAR) and showed that LinRace can improve upon the backbone tree by taking advantage of the gene expression data (Fig. 2e). Comparisons using other metrics are provided in Supplementary Fig. 5.

While using different backbone methods among these three choices (DCLEAR, Cassiopeia, and NJ) yields comparable results, the default combination we use (NJ and LinRace) performs slightly better than other choices. Moreover, we investigate how the methods perform with datasets of different numbers of cells, ranging from 128 cells to 2048 cells (Supplementary Fig. 6), with a fixed mutation rate of 0.1. As expected, the performances of all methods decrease with the increase in the number of cells, while LinRace outperforms all other methods.

The trends of the performance of all methods except for LinTI-MaT with the increase of mutation rate are expected and are consistent with the observation in ref. 15. For these methods, the optimal mutation rate for tree reconstruction is in the range of 0.03–0.05 with dropouts (Fig. 2a–d). Larger mutation rates cause more excision dropouts to occur, thus the quality of barcode data decreases, therefore, the performances decrease correspondingly. To quantify the quality of barcode data, we design a score named *reconstruction potential* ("Methods"). When calculating this, we make use of the known barcodes of ancestral cells from simulation and count how many edges that connect a parent cell with a daughter cell have at least one mutation introduced from the parent to the daughter. If there is at least one mutation on this edge, it means it is possible to reconstruct this edge of the tree (though it can still be very hard), which corresponds to a split of all leaf cells. If there is no mutation on the edge, we consider that the split cannot be reconstructed. Supplementary Fig. 3e shows the reconstruction potential of all datasets with 1024 cells with the change of mutation rates (blue boxes). The trends of the reconstruction potential with the mutation rates confirm the decreasing data quality as mutation rates become too large. Red boxes in Supplementary Fig. 3e show the RF distances of LinRace results. RF distance can be compared with reconstruction potential, as they are both based on the ratio of correctly reconstructed splits ("Methods"). The RF distance of LinRace can sometimes ($\mu = 0.1$ without dropouts) exceed the reconstruction potential, which can be attributed to refined local structures inferred from the gene expression data.

Overall, LinRace-TST and LinRace-IST show similar performances (Fig. 2). This similarity in performance indicates that LinRace is robust to inference errors in the cell states and cell state tree, which confirms the applicability of LinRace on real datasets, where dropouts are present and true cell states and cell state trees are not available. Since Slingshot was run on each dataset obtained with a given parameter setting and random seed separately, the inferred cell state tree for each dataset can be different, as visualized in Supplementary Fig. 7b. In addition, when inferring cell states, the ground truth number of cell states is usually unknown. In our evaluation, we used a generic parameter of 7 for the number of states for both simulated and the *C. elegans* real data ("Methods"). However, it should be noted that the ground truth number of cell states for the simulated dataset is 52 (Supplementary Fig. 7a). Despite this difference, our results demonstrate that LinRace is robust to the number of cell states, as evidenced by the comparable performances of LinRace-TST and LinRace-IST.

In the scenario where the barcode data quality is ideal, that is, with no dropouts and with optimal mutation rate (from 0.1 to 0.2 without dropouts), DCLEAR-kmer, which uses only barcode data, can achieve comparable (Supplementary Fig. 3) performance as LinRace, which means leveraging the gene expression data does not yield to significant improvement in this particular case. However, it is important to emphasize that the current lineage barcode data available is far from achieving such ideal quality. Therefore, incorporating gene expression data in a suitable manner becomes crucial for accurate lineage reconstruction.

## LinRace alleviates the barcode homoplasy problem

*Barcode homoplasy* refers to the occurrence of the same mutation to the same target site at multiple lineages in the cell division tree, which can cause cells from different clones to have the same barcode. This

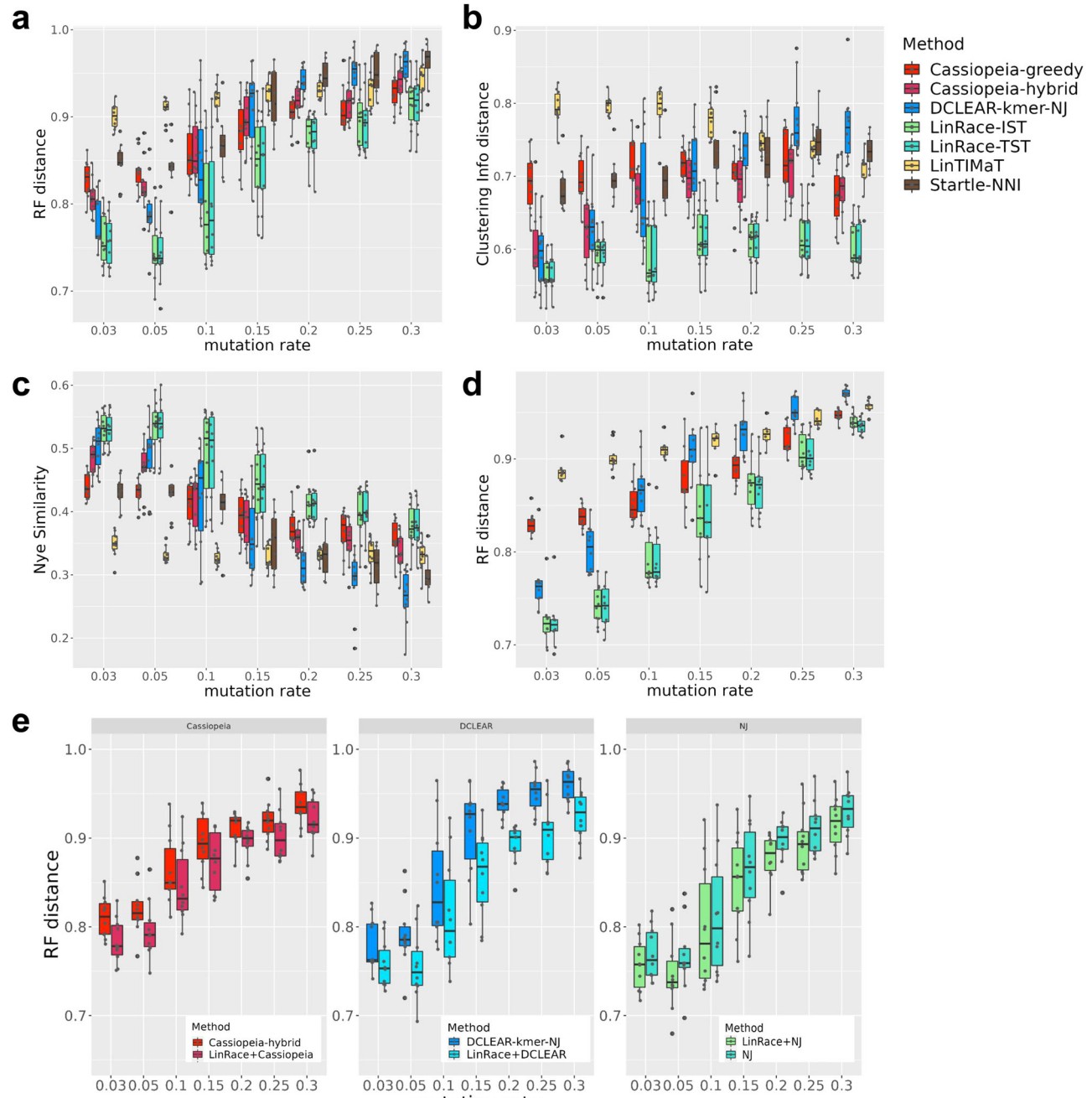

**Fig. 2 | Benchmarking results for lineage reconstruction methods on TedSim simulated datasets with dropouts.** For all boxplots shown in the figure, we adopt the following default settings in the ggplot2 library: center line, median; Upper and lower box limits, the 25th and 75th percentiles; whiskers, 1.5× interquartile range for both upper and lower ends. Source data are provided as a Source Data file.
**a–c** Comparisons of LinRace (LinRace-IST and LinRace-TST) and other methods on TedSim simulated datasets using RF distance, CID. and Nye similarity on datasets with 1024 cells. RF distance, Nye similarity, and CID all have the range of [0, 1]. For both RF distance and CID, lower is better, and for Nye similarity, higher values indicate better performance. The detailed descriptions for simulation and method settings can be found in "Methods" and Supplementary Note 2. **d** Comparisons of LinRace (LinRace-IST and LinRace-TST) and other methods on TedSim simulated datasets using RF distance on datasets with 4096 cells. Startle-NNI is excluded from this analysis due to computational time on this larger dataset. **e** RF distance comparisons of LinRace's improvements upon different methods for building tree backbone including NJ, Cassiopeia-hybrid, and DCLEAR-kmer. In these plots, we used the same datasets (with dropouts) as in (**a–c**). Comparisons using Nye Similarities and CID are included in Supplementary Fig. 5.

problem is worsened when: (1) certain mutations are highly frequent; and (2) there are dropouts on the barcodes. When using TedSim to generate simulated data in our tests, we model these two factors to generate realistic barcodes. As a result, we observe a large number of cells sharing the same barcode in both real and simulated data (Supplementary Fig. 1).

To verify that the frequency of the same mutation occurring on multiple lineages during the cell division process is high in our simulated data, we use a measurement named "homoplasy edge ratio" ("Methods"). The way we calculate the homoplasy edge ratio allows us to accommodate the dropout events which largely affect the barcode diversity. The homoplasy edge ratios of one of the datasets simulated by TedSim on 1024 cells and 16 characters are shown in Supplementary Fig. 8a. We can observe that when there are dropouts, the homoplasy edge ratio can reach as high as 0.75. Also, the ratio changes along the mutation rate with a similar trend as the performance of lineage

reconstruction methods as shown in Fig. 2a–d, indicating that the homoplasy edge ratio can reflect the barcode data quality.

Since LinRace refines the star trees of the cells with the same barcode using gene expression data, this strategy can be considered one way to target the homoplasy problem. While the superior performance of LinRace supports its effectiveness on datasets with popular barcode homoplasy (Fig. 2a–d), we here investigate an example from the benchmarks used in Fig. 2. For a group of cells sharing the same barcode but belonging to different clones in the true tree (Supplementary Fig. 8b), we extract their subtree topology by removing all excluded leaf cells as well as collapsing the internal node to obtain the true tree with only these cells (Supplementary Fig. 8c). Since their lineage barcodes are identical, barcode-based methods can only obtain a random tree (Supplementary Fig. 8e). The CID between the true tree and the LinRace tree/random tree showed that the LinRace reconstructed tree (Supplementary Fig. 8d) is superior to all 100 randomly generated trees by a large margin (Supplementary Fig. 8f). Supplementary Fig. 8g shows that the LinRace reconstructed tree is closer to the true tree than a random tree in terms of pairwise distance between cells on the respective tree.

Overall, barcode homoplasy presents a significant challenge for methods reconstructing cell division trees. The use of gene expression data in LinRace helps to partially recover the relative positions of cells with the same barcode in the tree. However, the gene expression information has limitations in fully resolving the homoplasy problem since the gene expression of cells is dominated by cell types rather than lineages. Moreover, for computational efficiency, LinRace runs tree refinement on local subtrees. To fully recover the positions of cells in the global tree, optimization on the global tree structure is needed, but methods that perform whole-tree structure optimization did not prove successful (e.g. LinTIMaT and Startle) due to the vast search space of the whole tree.

## Computational efficiency analysis of LinRace

The computational efficiency of tree reconstruction algorithms is an important aspect to consider. The computational cost of calculating the likelihood as well as the size of the tree space increases super-exponentially with the number of leaf cells. Therefore, most existing methods such as DCLEAR and Cassiopeia use greedy heuristics to enable efficient tree reconstruction.

In LinRace, we perform local search to learn subtrees on the cells with exactly the same barcode, therefore, for each dataset, the size of the subtrees LinRace needs to learn can vary for every unique barcode. The number of iterations is one of the key parameters for local search algorithms, and often, more iterations are needed for larger trees. We adopt a dynamic manner to set the number of iterations for each subtree based on the Catalan number[33], such that smaller subtrees use fewer iterations than larger subtrees, to improve the efficiency of the algorithm.

Supplementary Fig. 9 shows the comparison of LinRace with other state-of-the-art methods in terms of running time on different numbers of input cells. LinRace runs much faster than LinTIMaT, which is the other integrated method that uses both barcode and gene expression data. Barcode-only tree reconstruction methods, Cassiopeia-greedy and DCLEAR, LinRace has a similar running time when the number of cells is small, and its running time grows faster than these two methods when the number of cells increases, in order to search for subtrees that maximize the likelihood function. LinTIMaT has the largest running time. Startle-NNI is not included in this analysis due to not closing the program properly.

## Evaluating LinRace and baseline methods on real *C. elegans* dataset

The ideal scenario to evaluate lineage tree reconstruction methods is to have the following information: the gene expression data and

barcode data for single cells, and the ground truth cell lineage tree. While it is rare for experimental data to have ground truth cell lineage trees, *Caenorhabditis elegans* (*C. elegans*) is one of the few species that have the exact cell lineage resolved[34] with single-cell gene expression data measured. To obtain paired single-cell gene expression and barcode data for this system, we adopt a strategy used in ref. 22 to simulate the barcode data from the known lineage tree. Therefore, we combine the measured gene expression data of *C. elegans* and simulated lineage barcode data from TedSim using the true lineage tree. We then apply different lineage reconstruction methods and compare the reconstructed lineage trees with the ground truth tree. When simulating the barcode data, we again vary the mutation rate and the existence of dropouts.

The dataset is obtained from ref. 35, who profiled the gene expression lineage of 93 genes in 363 specific cells from L1 stage larvae. They used knowledge of the cell number, morphology of the cell nuclei, and their relative position with respect to each other to develop an automatic method to identify specific cells in confocal images of worms expressing a fluorescent reporter, and then measure expression in specific cell nuclei. The Newick format of the true lineage of the L1 larvae is obtained from CeLaVi[15] by Salvador-Martinez et al. which is then trimmed to the profiled 363 cells in the dataset (see Fig. 3a). From the visualization of reduced dimensions in Fig. 3b, we can see that the inferred trajectory is able to connect the cell states and form a continuous manifold. The inferred cell state tree (shown in Supplementary Note 2) from Slingshot[28] is used to calculate the state transition likelihood in LinRace.

From the results in Fig. 3c and Supplementary Fig. 10, we can see that LinRace outperforms the state-of-the-art methods consistently for varying mutation rates, with (Fig. 3c) or without (Supplementary Fig. 10) dropouts. The overall results are consistent with the evaluation results on simulated datasets, which also confirms the benchmarking capability of the simulated datasets to a certain extent. These results not only show the advantages of LinRace over baseline methods on real data but also confirm that LinRace does not need a true cell state tree to be effective. As long as the trajectory inference method captures the relative local relations between cell states, our likelihood function can effectively evaluate a candidate lineage tree based on the raw expression and state transitions. Comparing the performances of the methods with and without dropouts, we can see that LinRace outperforms other methods even more on datasets with dropouts, which indicates that integrated methods like LinRace are able to utilize the gene expression data to compensate for the loss of information caused by dropouts in barcode data. On the other hand, despite using gene expression data, LinTIMaT does not perform better than the barcode-based methods with dropouts, DCLEAR, Startle, and Cassiopeia. The reason can be twofold: first, LinTIMaT runs local search on the whole lineage tree which allows it to explore only a small proportion of the search space, as optimizing this tree as a whole is an NP-hard problem[36]; second, the design of their likelihood function is based on an over-simplified assumption on the relationship between gene expression and barcode data.

## LinRace reveals ancestral state transitions of zebrafish brain cells

In most studies that present jointly profiled scRNA-seq and lineage barcode data[3–6,8–11], the barcode data and the gene expression data are processed and analyzed separately, where the barcode data is used for building the cell lineage and the gene expression data is used to infer the cell types (Supplementary Fig. 11a). Due to the poor quality of the barcode data, the lineage tree tends to have a relatively low resolution, which is reflected by the shallow depth and the small number of internal nodes. Hundreds of cells of different cell types can be connected to the same node and their relative clonal relationships are unknown.

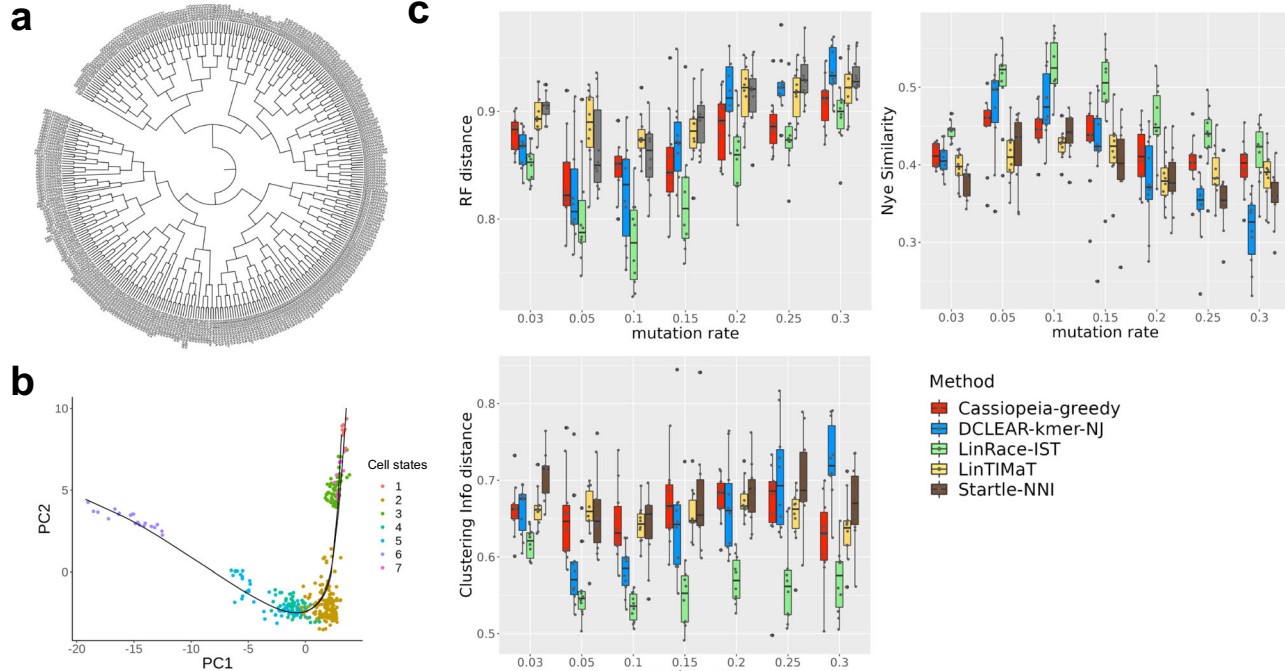

**Fig. 3 | Results of tree reconstruction methods on real *C. elegans* datasets.** Source data are provided as a Source Data file. **a** The ground truth lineage of *C. elegans* at L1 larvae stage. There are 363 profiled cells on the tree, while the original L1 larvae lineage has 668 cells. The tree pruning is performed using the `ape` package in R and the clonal relations between cells are preserved. The tree is visualized using iTOL[42]. **b** 2-D PCA visualization of the gene expression of the *C. elegans* data. After PCA, the first 20 PCs are used for K-means clustering and Slingshot to determine cell states and state trajectories (cell state tree). A detailed description of the processing steps for the gene expression data can be found in Supplementary Note 2. **c** Evaluating result on the *C. elegans* dataset with simulated barcodes. The methods are tested for varying mutation rates with dropout effects. Three metrics, RF distance (lower is better), Nye similarity (higher is better), and CID (lower is better) are presented. For all boxplots shown in the figure, we adopt the following default settings in the ggplot2 library: center line, median; Upper and lower box limits, the 25th and 75th percentiles; whiskers, 1.5× interquartile range for both upper and lower ends.

In this section, we show that LinRace can be used to obtain cell lineage trees with better local resolution than the state-of-the-art methods, as well as the cell states of ancestral cells, which is a unique function of LinRace. To infer the cell states of ancestral cells, we assume that all cell states in the developmental process are captured in the scRNA-seq data and included in the cell state tree. The finer local structure of the inferred lineage tree inferred by LinRace together with the cell states allows us to identify the location of symmetric and asymmetric divisions in the reconstructed lineage and obtain a picture of how cell types are formed during cell divisions.

We used a zebrafish brain dataset from scGESTALT[9] and reconstructed the cell lineage using LinRace and other baseline methods. When running LinRace, we used inferred cell states from *k*-means and inferred cell state tree from Slingshot (Supplementary Fig. 11b). The reconstructed cell division trees are visualized in Fig. 4a and Supplementary Fig. 12. We are not able to provide quantitative accuracy of the reconstructed trees as there does not exist a ground true lineage tree. Instead, we compare the resolutions of the trees using the depth and number of internal nodes of the reconstructed trees. From Fig. 4b, we see that the LinRace reconstructed lineage tree has more depth and inferred internal nodes than the Cassiopeia tree and LinTIMaT tree. For DCLEAR which also infers a binary tree, the local splits between cells with the same barcode are randomly decided which does not provide any biological insights, and its number of internal nodes is almost the same as that of LinRace, but its depth is much higher means that the DCLEAR tree is much less balanced than the LinRace tree (Supplementary Fig. 12).

Figure 4c shows two Gene Expression Subtrees (GES) reconstructed by LinRace, where the color of nodes (including both leaf nodes and ancestral nodes) represent the cell types of cells, as annotated in the original paper[9]. Although LinRace used inferred cell states and the cell state tree when performing tree reconstruction, we analyzed the reconstructed lineages by labeling the cells with annotated cell types. To obtain the states of ancestral cells, we ran the ancestral state inference procedure ("Methods") again using the annotated cell types of leaf cells and a cell state tree of these cell types (Supplementary Note 2). After obtaining the cell states of both leaf and ancestral cells, we can examine the cell state changes on the reconstructed lineages, and observe how the progenitor cells adopt asymmetric divisions to generate different clones of Fore/MidBrain cell types. The two GESs are dominated by Forebrain and Midbrain cell types, respectively (Fig. 4c, left and middle). In each GES, the cells have the same barcode, so without using gene expression information, no meaningful structure can be reconstructed for each GES. LinRace reconstructs these subtrees along with the states of ancestral cells. From Fig. 4c, we can see where some progenitors undergo asymmetric divisions and differentiate into various cell types after self-renewal (Subtree 1 in Fig. 4c, left), while cells that have entered a Forebrain cell type divide mostly symmetrically to maintain the sustainable activity of neurogenesis (Subtree 2 in Fig. 4c, left). In Fig. 4c, middle, multiple symmetric divisions happen to lead to the Midbrain cell types.

Finally, from the visualization of reconstructed cell lineage trees with different methods (Fig. 4a and Supplementary Fig. 12), we can see that compared to the other methods, LinRace is able to reconstruct a lineage tree with a realistic distribution of cell types, where while cells with the same cell types tend to be located together in the reconstructed lineage tree, the same cell type can appear in different subtrees, shows the "partial consistency between transcriptome similarity and barcode similarity" in real data. In the next section, we further show this phenomenon using another dataset of the mouse embryo. Further, we show how asymmetric divisions explain this phenomenon

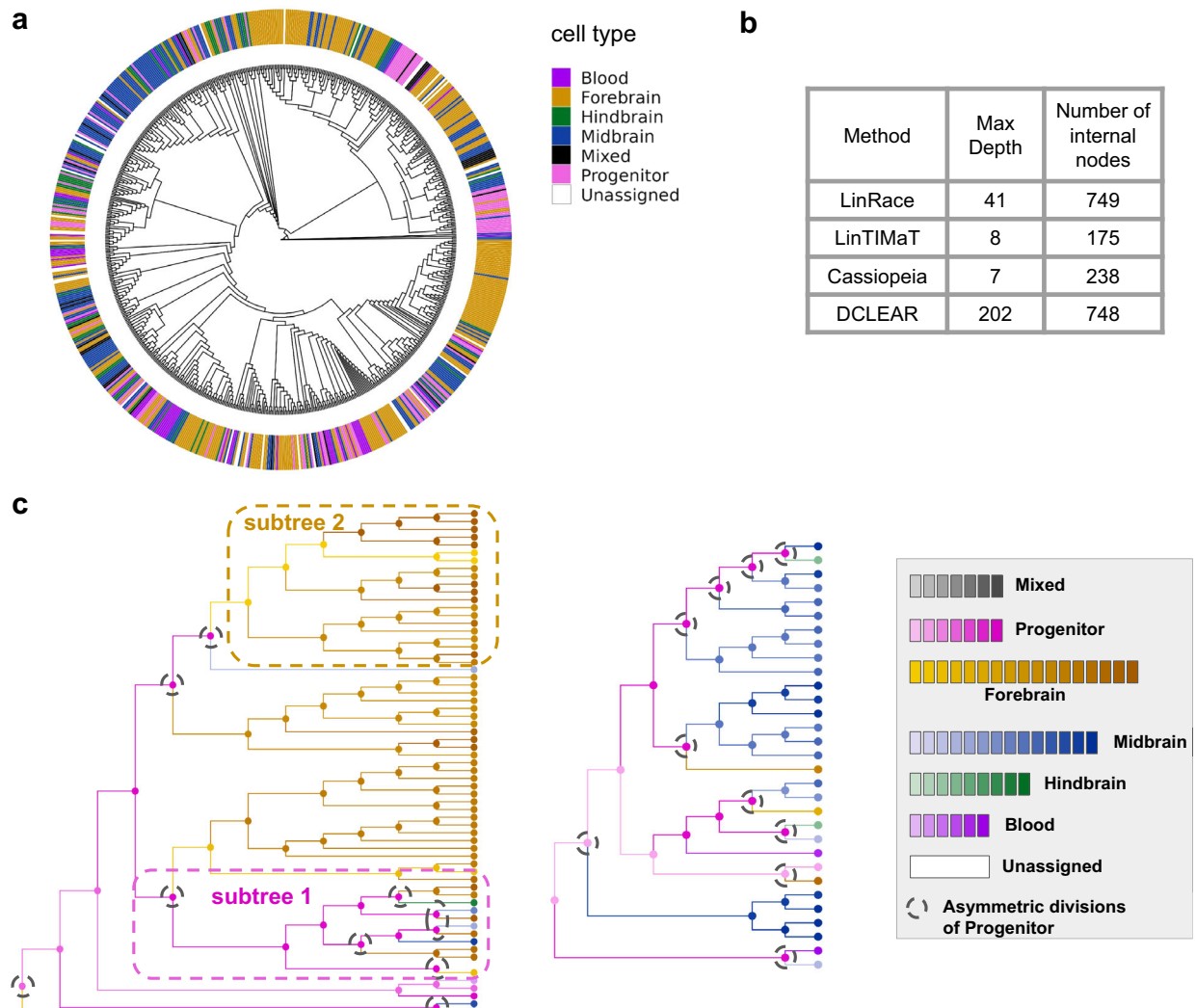

**b**

| Method | Max Depth | Number of internal nodes |
|---|---|---|
| LinRace | 41 | 749 |
| LinTIMaT | 8 | 175 |
| Cassiopeia | 7 | 238 |
| DCLEAR | 202 | 748 |

**Fig. 4 | Reconstructed trees of ZF1-F3 sample (750 cells) from scGESTALT datasets in[9].** Source data are provided as a Source Data file. **a** LinRace reconstructed tree. The colors of the outer ring represent major cell-type assignments. **b** Properties of the reconstructed trees from different methods. The max depth means the maximum total edge length going from the root to any leaf cell. **c** A detailed look at two GES (Gene Expression Subtrees) in LinRace reconstructed tree with inferred ancestral states. In the left GES, Subtree 1 shows a subtree of progenitors' self-renewal, and Subtree 2 shows a subtree of differentiated Forebrain population. Black dashed circles denote asymmetric divisions of progenitor cells. The annotated leaf cell states are from the original data paper, and the ancestral states of all hidden nodes are inferred using LinRace. Detailed descriptions of the processing steps for the gene expression data are in Supplementary Note 2.

and the conjecture of varying cell differentiation speeds on different lineages.

### LinRace helps to answer the sources of diverse cell types in the mouse embryo data

We applied LinRace to an early mouse embryo dataset from ref. 5 to infer the cell division tree. The mouse embryo dataset contains 9707 cells of 34 annotated cell types from the authors (Fig. 5a). On this dataset, we not only analyze how the cell types are distributed over the cell division tree but also ask if any lineage signature exists in a cell's gene expression profile.

We first cut the reconstructed tree at depth 26 and obtained subtrees with their roots at a distance of 26 to the root of the complete tree (Fig. 5b). First, we investigated the cell-type composition of the subtrees. The cell-type composition of seven subtrees with at least ten leaves is shown in Fig. 5c. Each subtree consists of multiple cell types (colors are consistent with the color legend of Fig. 5a), and the same cell type appears in different subtrees, again showing the "partial consistency between transcriptome similarity and barcode similarity".

Next, we take the cells that belong to the same cell type ("Fore/Midbrain" in Fig. 5a) and look for differences that potentially exist between cells from different lineages but with the same cell type. We visualize the Fore/Midbrain cells labeled by their clonal IDs (same as the subtree IDs, Fig. 5d). In the UMAP space, cells from different clones are mixed and do not show any cluster pattern. Differential expression (DE) analysis between cells in different subtrees and the same cell type did not show DE genes whose functions can clearly cause clonal differences. Further, we took three relatively large clones of the cells in the Fore/Midbrain cell types, the clones 9, 10, and 12 as labeled in Fig. 5d, and calculated intra-clone and inter-clone gene expression distances among these clones (Fig. 5e, "Methods"). As shown in Fig. 5e, there are no observable differences between the intra-clone distances and the inter-clone distances. These observations indicate that there are no significant lineage-affiliated features in the reduced dimensions of the transcriptomic data, which is in line with the hypothesis that the gene expressions of mature cell types are dominated by their functionality, not their lineage identity, as previously reported in ref. 34.

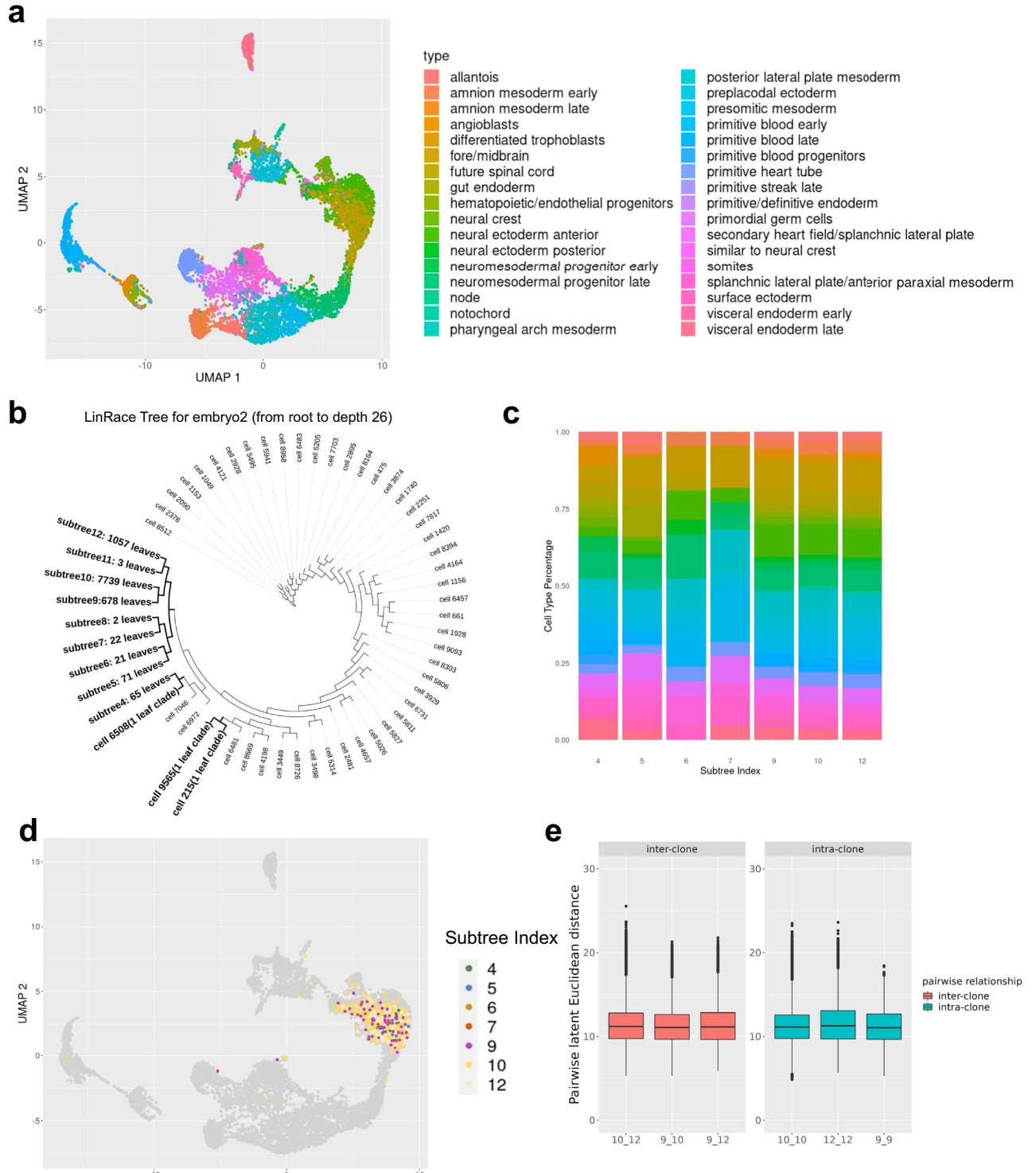

**Fig. 5 | Visualizing inconsistencies between cell states and lineages. a** UMAP visualization of the early mouse embryo dataset with cell-type annotation from ref. 5. **b** LinRace reconstructed tree of the dataset. Only branches from root to *depth* = 26 are shown here, where the remaining subtrees are shown as leaf nodes, with the numbers of leaf cells attached to them. Three one-leaf clades are included, which means they are leaf nodes at *depth* = 26. Source data are provided as a Source Data file. **c** Cell-type compositions under different subtrees. The subtrees are obtained by cutting the whole lineage at the same depth (total edge length from the root to the cutting point.) 12 subtrees are obtained by cutting the lineage tree at *depth* = 26 and only subtrees have more than ten leaves are shown here. The *y* axis shows the percentage of each cell type in (**a**) under each subtree. **d** UMAP

visualization of the Fore/Midbrain cell type by the clonal IDs. Only subtrees that have more than 20 leaves are labeled, and cells that do not belong to these subtrees are colored gray in the background. **e** Intra-clone and inter-clone distances between cells based on their expression profiles. Cells from clones 9, 10, and 12 in the Fore/ Midbrain cell type are involved in this comparison. For every pair of cells, the Euclidean distance is calculated on the first 30 PCs after PCA. The cell-cell distances are grouped into intra-clone and inter-clone categories. In this figure, we adopt the following default settings in the ggplot2 library: center line, median; Upper and lower box limits, the 25th and 75th percentiles; whiskers, 1.5× interquartile range for both upper and lower ends.

## Discussion

In this paper, we present LinRace, an integrated method that combines the lineage barcode and gene expression using the asymmetric cell division model to reconstruct cell division trees. Compared to the state-of-the-art methods, LinRace has the following advantages: (1) Using the combined framework of NJ and maximum likelihood, LinRace outperforms the state-of-the-art methods on both simulated and real datasets; (2) LinRace proposes a likelihood function that takes into account mRNA counts, cell states and state transitions to find the lineage tree that best explains the observed gene expression data; (3) LinRace is able to infer the cell states of ancestral and provides insights on the process of generating new cell types; (4) LinRace performs a local search on subtrees using a dynamic number of iterations based on the size of the subtrees which makes it computationally efficient compared to other tree search algorithms.

Reconstructing the cell division tree from lineage tracing barcodes is similar to the problem of inferring phylogenetic trees from genome data in the field of evolutionary biology[37]. However, the cell division tree reconstruction is even more challenging due to the much larger number of leaves (single cells) and the short lineage tracing barcodes with dropouts. Incorporating the single-cell gene expression data is a promising direction, but developing such integrated methods requires modeling the relationships between the two modalities, the lineage barcode, and the gene expression. LinRace adopts the asymmetric division model, which explains phenomena in real data, increases reconstruction accuracy, and allows for the estimation of ancestral cell states.

LinRace assumes that cell state transitions are irreversible, which is generally true under homeostatic conditions. However, reversible state transitions are also known to exist. To incorporate reversible transitions, it may be necessary to expand the definition of the cell state tree to a cell state network in future work. Another assumption we made, is that all cell states during the developmental process exist in the current-day cells. This is also the assumption made by trajectory inference methods in general. For certain developmental systems where cell states change fast, certain ancestral or intermediate cell states cannot be captured in the scRNA-seq data. If LinRace is unaware of these cell states, the cell states it reconstructs for ancestral cells will be limited only to the cell states that are present in the scRNA-seq data. Therefore, LinRace is most applicable to systems with asynchronous differentiation, where most of the ancestral cell states exist in current-day cells. However, if time series scRNA-seq data exist for the system under study, such data can allow for the reconstruction of a comprehensive cell state tree that includes ancestral states of the paired scRNA-seq and lineage barcode data. In this case, one can simply use the cell state tree inferred from the time series scRNA-seq data. In addition to developing new computational methods, it is also important that the data quality is improved with advances in technology, e.g., to create lineage barcodes with high diversity and coverage[7], in order to build the tree of cell division and cell differentiation with high accuracy.

## Methods

### The asymmetric division model accounts for cell state changes in the lineage tree

The asymmetric division of cells is a key process that leads to multiple cell types and different cell differentiation speeds on different clones on the cell lineage tree[38,26]. We show that the asymmetric division model can account for various scenarios of cell state change on a cell division tree reviewed in ref. [2]. Moreover, it was shown to lead to realistic paired lineage barcode and gene expression data[14]. An asymmetric division results in two unequal daughter cells from the parent cell, one of which differentiates into a natural "next-state" (according to the cell state tree) while the other remains in the same cell state as the parent.

Wagner and Klein[2] reviewed hypothetical scenarios of restricted lineage trajectories unfolding on a "state manifold", or in our terms, the developmental trajectories of gene expressions. In the review, state convergence represents two or more distinct fate trajectories converging onto the same final position on a state manifold, and state divergence represents the reverse process where one trajectory bifurcates into two or more distinct fate trajectories. These seemingly contradictory scenarios can happen due to asymmetric cell divisions. Asymmetric divisions can cause one cell state to generate two distinct cell states, resulting in state divergence; and cells on distant lineage trees can also divide asymmetrically into the same cell state, resulting in state convergence. In LinRace, we account for both symmetric and asymmetric cell divisions in our likelihood function using the asymmetric division likelihood, which estimates the prior probability of asymmetric and symmetric divisions based on the mutated states in the observed cells. We assume that state transitions on all parent-child cell edges are independent and also consider the stochasticity of cells' developmental speeds by varying the number of states traversed for each state transition. The symmetric and asymmetric cell divisions that happen on the cell lineage tree according to certain probabilities are called the asymmetric division model in this paper. In LinRace, the asymmetric division model is also used in inferring ancestral cell states.

### Reconstructing lineage backbone from the lineage barcode data

The lineage barcode of a cell is represented as a character vector of length equal to the number of target sites as designed by the CRISPR/Cas9 lineage recorder. Each character represents a state of the target site, which can be a mutated state, an unmutated state, or a dropout state. We use "0" to represent the unmutated state, and each unique non-zero character represents a unique mutation state, regardless of the position where the mutation is observed. The dropout state is denoted as the "−" character. For the barcode data, we assume that at the root of the cell lineage, an unedited DNA sequence (all unmutated states) is introduced. During cell divisions, unmutated states can potentially mutate and will never mutate again, except the dropout state. Given $N$ cells and $M$ targets, the lineage barcode data is a $N \times M$ matrix. For the barcode data, we assume that at the root of the cell lineage, an unedited DNA sequence is introduced. During cell divisions, unmutated target sites can potentially mutate and will never mutate back to the original state (except dropout state).

In LinRace, we utilize a Hamming distance-based NJ method to infer the "backbone lineage tree". Since multiple cells can have the same barcode in the dataset, running Neighbor Joining on the $N \times M$ matrix will result in merging cells with the same barcode in some random order. With a given lineage barcode matrix, we first transform it into a $K \times M$ matrix where each row represents a unique barcode in the original data, and then Neighbor Joining is applied to the $K \times M$ matrix to get the lineage tree of unique barcodes (Fig. 1). We call this tree the reconstructed lineage backbone, or tree backbone (denoted by $\mathcal{T}_0$).

### Inferring cell states for ancestral cells

To calculate the likelihood of a candidate tree based on the gene expression profiles of the leaves, we first need to infer the states of cells at ancestral nodes using the states of leaf nodes and the cell state tree that are inferred during Fig. 1, Step B. The inference of ancestral cell state follows the rule of asymmetric division. From leaves to the root of the candidate lineage tree, we consider the following cases: (1) If the daughter cells have the same cell state, their parent is assigned the same state, meaning the parent cell divides symmetrically. (2) If the daughter cells have different cell states, the parent cell will have the Most Recent Common Ancestor (MRCA) cell state based on the cell state tree. The ancestral cell states in trees in Fig. 1, Step C follow these two rules given the cell state tree learned from Step B. This ancestral

state inference process allows one cell to divide into two cells with different cell states. If two daughter cell states belong to the same differentiation path (a path from the root state to a leaf state), the parent cell will be in the same cell state as the earlier cell state between the two.

### Finding subtree topology using a maximum-likelihood method
Cells with identical barcodes form a star tree in the initial NJ tree $\mathcal{T}_0$ (see Fig. 1). LinRace utilizes the gene expression data to learn the binary tree topology (every non-leaf node has exactly two child nodes) of these cells. The learned binary trees, termed Gene Expression Subtrees (GES), are then attached to the tree backbone to yield the full cell lineage tree. We design a maximum-likelihood scoring function and local search strategy to find the GES.

**Likelihood of a candidate lineage tree.** We design a likelihood function to evaluate how well a candidate lineage tree explains the observed gene expression data. We use the ancestral state inference step mentioned above to determine the cell states of all nodes on the tree. Then, we can calculate the likelihood of the tree, which consists of three terms: *state transition likelihood*, *asymmetric division likelihood*, and *neighbor distance likelihood*. The first two terms use the cell state information of cells, and the last uses the gene expression profiles of the cells.

The *state transition likelihood* represents the probability of transitions between cell states on the edges of the lineage tree. We adopt the assumption that the state transition on each edge is independent of other transition events (this assumption is commonly used in phylogenetic tree reconstruction) so that we can write the state transition likelihood of a given lineage tree $\mathcal{T} = (E, V)$ as:

$$\mathcal{L}_{\mathcal{E}}(\mathcal{T}) = \log P(\mathcal{S}|\mathcal{T}) = \log \prod_{e \in E} P(\mathcal{S}_{e_2}|\mathcal{S}_{e_1}) = \sum_{e \in E} \log P(\mathcal{S}_{e_2}|\mathcal{S}_{e_1}) \tag{1}$$

where $e = (e_1, e_2)$ represents an edge on the tree graph from node $e_1$ to $e_2$, $\mathcal{S}$ denotes the cell state assignments of all cells, and $\mathcal{S}_{e_1}$ and $\mathcal{S}_{e_2}$ denotes the cell states of the two cells $e_1$ and $e_2$ connected by $e \in E$.

The state transitions of cells' gene expressions are governed by an underlying developmental cell state tree, which is inferred from the gene expression data using Slingshot (Fig. 1, Step B). The cell state tree guides the cells to differentiate into certain future cell states irreversibly. For any two states on the cell state tree, there can exist at most one path (a sequence of connected, directed edges) that links these two states. Therefore, when the states of a pair of ancestor-descendant, denoted as $(\mathcal{S}_{e_1}, \mathcal{S}_{e_2})$ between two cells $(e_1, e_2)$, the transfer probability $P(\mathcal{S}_{e_2}|\mathcal{S}_{e_1})$ is calculated as follows:

$$P(\mathcal{S}_{e_2}|\mathcal{S}_{e_1}) = P(\mathcal{D}(\mathcal{S}_{e_1}, \mathcal{S}_{e_2})) \tag{2}$$

where $\mathcal{D}(\mathcal{S}_{e_1}, \mathcal{S}_{e_2})$ represents the graph geodesic from $\mathcal{S}_{e_1}$ to $\mathcal{S}_{e_2}$ on the cell state tree. If there is no path between the two states on the cell state tree, the graph geodesic is defined as infinity: $\mathcal{D}(\mathcal{S}_{e_1}, \mathcal{S}_{e_2}) = +\inf$. We define the subset of edges with non-infinity-state graph geodesic as:

$$E_{state} = \{e = (e_1, e_2) \in E \mid \mathcal{D}(\mathcal{S}_{e_1}, \mathcal{S}_{e_2}) \neq +\infty\}$$

Then, we infer the probability for every distinct state transition as follows:

$$P(\mathcal{S}_v|\mathcal{S}_u) = \frac{\sum_{(u',v') \in E_{state}} \mathbb{1}(\mathcal{D}(\mathcal{S}_{u'}, \mathcal{S}_{v'}) = \mathcal{D}(\mathcal{S}_u, \mathcal{S}_v))}{|E_{state}|} \tag{3}$$

where $\mathbb{1}(x)$ is the characteristic function which returns 0 if $x$ is TRUE; 0 otherwise. For the case where the graph geodesic between two states is

infinity, which in turn makes $P(\mathcal{S}_v|\mathcal{S}_u)$ to be zero, we add a −50 penalty to the log-likelihood.

The *asymmetric division likelihood* considers the asymmetric divisions in the lineage tree, and it is defined as follows:

$$\mathcal{L}_{ad}(\mathcal{T}) = \sum_{(s_{ances}, s_1), (s_{ances}, s_2) \in \mathcal{T}} \log P_{ad}(s_1 = s_2) \tag{4}$$

where

$$P_{ad}(s_1 = s_2) = \begin{cases} 1 - p_a, & s_1 = s_2 \\ p_a, & s_1 \neq s_2 \end{cases} \tag{5}$$

where $s_{ances}$ is the state of a parent cell and $s_1, s_2$ are the states of the two descendant cells. The asymmetric division rate $p_a$ can be determined using prior knowledge or inferred from the fraction of observed asymmetric neighbors in the lineage tree.

For *neighbor distance likelihood*, we look at cells that are siblings (having the same parent node) on the candidate lineage tree, and use the transition probability from diffusion map[39] to evaluate if the two cells are locally connected on the developmental trajectories of the gene expression data. Even though in general when asymmetric division happens, the two daughter cells do not have the same cell states thus their transcriptomes are not very similar, we consider that many cells at the leaves of the lineage tree are leaf state cells, and asymmetric divisions are less prominent when more cells are at leaf states. We denote the set of measured cells' gene expression by $\Omega$. First, for each pair of cells $(i, j)$, denoting their expression vectors to be $\mathbf{x_i}$ and $\mathbf{x_j}$ respectively, where $\mathbf{x_i}, \mathbf{x_j} \in \Omega$, we calculate the pairwise distance using the radial basis function (RBF) kernel $K_{ij} = exp(\frac{-||\mathbf{x_i} - \mathbf{x_j}||^2}{2\sigma^2})$. Then, the transition probability between any two cells can be calculated as follows:

$$Z(\mathbf{x_i}) = \sum_{\mathbf{x_k} \in \Omega} K_{ik} \qquad \hat{Z}(\mathbf{x_i}) = \sum_{\mathbf{x_k} \in \Omega/\mathbf{x_i}} \frac{K_{ik}}{Z(\mathbf{x_i})Z(\mathbf{x_k})} \tag{6}$$

$$P_{nd}(\mathbf{x_i}, \mathbf{x_j}) = \frac{1}{\hat{Z}(\mathbf{x_i})} \frac{K_{ij}}{Z(\mathbf{x_i})Z(\mathbf{x_j})} \tag{7}$$

similarly, the neighbor distance likelihood can be calculated as:

$$\mathcal{L}_{nd}(\mathcal{T}) = \sum_{\mathbf{x_i}, \mathbf{x_j} \in \Omega, (\mathbf{x_i} \leftarrow u \rightarrow \mathbf{x_j} \in \mathcal{T})} \log P_{nd}(\mathbf{x_i}, \mathbf{x_j}) \tag{8}$$

Finally, the total likelihood is calculated as:

$$\mathcal{L}(\mathcal{T}) = \mathcal{L}_{\mathcal{E}}(\mathcal{T}) + \lambda_1 \cdot \mathcal{L}_{ad}(\mathcal{T}) + \lambda_2 \cdot \mathcal{L}_{nd}(\mathcal{T}) \tag{9}$$

where $\lambda_1$ and $\lambda_2$ are hyperparameters.

To find the best GES based on our likelihood function, we utilize hill-climbing local search to search in the tree space (Fig. 1, Step C). In order to propose a new tree from a current tree, we adopted random Subtree Swapping (rSS), which is a derivative of the widely used random Subtree Pruning and Regrafting (rSPR)[40]. We randomly select two nodes on the current tree and prune the subtrees attached to the specific nodes. Then, we regraft either one of the pruned subtrees to the location of the other subtree. The advantage of rSS is that with an initialization of a binary tree, this operation will not disrupt the binary property throughout the searching process. This move can result in most of the applicable topological changes of the tree, so we use it as the search technique. At every iteration, a new tree is proposed, which is one rSS move away from the current tree. Then, we evaluate the new tree with the likelihood function and compare it with the likelihood of the current tree. If the new tree has a higher likelihood, we move from the current tree to the new one. Repeat the process for every iteration

until no better tree can be found within a number of iterations, and the current tree is identified as a local optimal tree. We use a random restart technique after a local optimal tree is found, and try to find as many local optimal as possible within the number of maximum iterations. If multiple local optima are found, we return the tree with the highest likelihood as the optimal GES. The pseudocode of the local search process of LinRace is given in Supplementary Note 1.

## Simulating synthetic datasets with TedSim

We use simulated datasets from TedSim which generates paired lineage barcode and gene expression data with ground truth information of the lineage tree. Using the simulated datasets, we are able to benchmark LinRace and other tree reconstruction methods by comparing the reconstructed trees with the ground truth. To simulate with TedSim and test the methods' potential under various conditions. We use a pre-defined cell state tree (Supplementary Fig. 7a) and vary key parameters including mutation rate and the presence of dropout to simulate lineage barcodes of different quality. We used the "biased mode" of TedSim, where barcode mutations do not have the same probability to occur, which is the case in real data, thus simulating the realistic distribution of the mutations in the lineage barcodes. This mode can lead to a large number of cells sharing the same barcode in the data, as presented in Supplementary Fig. 1. The selected mutation rate ranges from 0.03 to 0.3 per target per cell division, which covers the realistic ranges of mutation rate in real datasets.

Another important advantage of TedSim is that TedSim simulates realistic dropout effects that widely occur in real datasets. In TedSim, when two or more mutation happens at the same cell division, the excision dropout will happen, resulting in deletions of the targets in between. These dropouts can cause barcode homoplasy, which leads to a significant decrease in barcode diversity. Comparing the distributions of unique barcodes of real dataset[5] (Supplementary Fig. 1), a simulated dataset with dropout, and a simulated dataset without dropout, we can see that TedSim simulation with dropout is able to generate realistic data with similar negative exponential distribution. In the real dataset of 19,019 cells and 2788 unique barcodes, and as mentioned in Results, 1929 of the 2788 barcodes uniquely label one cell; 330 of the 2788 barcodes are shared by two cells. In a simulated dataset of 1024 cells with dropout, 88 cells are uniquely barcoded (no other cell shares the same barcode) and 44 barcodes are shared for 2 cells. This means, 82.8% of the cells share the same barcode with at least two other cells, which is similar to the proportion of 86.4% in the embryo2 dataset in the mouse lineage tracing system. Without dropout, the number of uniquely barcoded cells increases to 336 and 136 barcodes are shared for 2 cells. The proportion of cells that share the same barcode with at least two other cells decreases to 40.6%. TedSim-generated datasets with dropouts reflect the distribution of the number of cells using the same barcode reflect the distribution in real data (Supplementary Fig. 1).

The cell state tree and cell states used for simulation are in Supplementary Note 2. The step size parameter in TedSim is used to sample the cell state tree to obtain discretized cell states from the tree. The step size in our experiments is set to be 0.5 which yields 52 discrete cell states in the simulated datasets. Other static simulation parameters can also be found in Supplementary Note 2.

In the experiment with the *C. elegans* dataset, TedSim is used to simulate lineage barcodes on the ground truth lineage tree. We also tune the mutation rate from 0.03 to 0.3 per target per cell division and simulate lineage barcode data with and without dropouts. We choose a smaller number of target sites ($Nchar = 9$) considering the size of the lineage (363 cells).

## Evaluation metrics for tree reconstruction

To evaluate the accuracy of reconstructed lineage trees, we use three tree comparison metrics: RF distance, Nye similarity, and CID

(Clustering Information Distance) distance. We use the RF.dist() function from the `phangorn` package to calculate the RF distance between reconstructed lineage trees and ground truth trees from simulated datasets. The function returns the ratio of inconsistent splits between the two trees, where a split is defined as the two sets of leaf nodes separated by an edge. Therefore, the distance value ranges from 0 and 1, where 0 represents a perfect reconstructed tree and 1 means the worst reconstructed trees where all splits are different from the true tree. When comparing two splits, one from the reconstructed tree and the other from the true tree, they are considered "consistent" only if the partitions of leaves indicated by the splits are exactly the same. This makes the RF distance a strict measure which can also be biased towards certain tree topology.

The second metric we use is the Nye Similarity score[31], which can be a complementary metric of RF distance. Like RF distance, Nye Similarity also compares the split corresponding to the edges on the two trees (the reconstructed and the true trees), but the result of comparing two splits (two edges) is not simply "consistent" and "inconsistent" as used in RF distance, and instead, a score based on the similarity between the two splits is calculated. When comparing edge $e_i$ in $\mathcal{T}_1$ ($\mathcal{T}_1$ is the reconstructed tree) with edge $e_j$ in $\mathcal{T}_2$ ($\mathcal{T}_2$ is the true tree), the score $s(e_i, e_j)$ is obtained by the comparing the partition of the leaf nodes corresponding to $e_i$ in $\mathcal{T}_1$ and the partition of the leaf nodes corresponding to $e_j$ in $\mathcal{T}_2$. Considering the set of all leaf nodes as $\mathcal{L}$, we have:

$$P_{e_{il}} \cup P_{e_{ir}} = P_{e_{jl}} \cup P_{e_{jr}} = \mathcal{L} \tag{10}$$

where $P_{e_{il}}, P_{e_{ir}}$ are the two disjoint subsets of $\mathcal{L}$ by the edge $e_i$, and $P_{e_{jl}}, P_{e_{jr}}$ are the two disjoint subsets by the edge $e_j$. Then, we calculate the following quantities,

$$a_{ll} = \frac{|P_{e_{il}} \cap P_{e_{jl}}|}{|P_{e_{il}} \cup P_{e_{jl}}|}, \ a_{rr} = \frac{|P_{e_{ir}} \cap P_{e_{jr}}|}{|P_{e_{ir}} \cup P_{e_{jr}}|}, \ a_{lr} = \frac{|P_{e_{il}} \cap P_{e_{jr}}|}{|P_{e_{il}} \cup P_{e_{jr}}|}, \ a_{rl} = \frac{|P_{e_{ir}} \cap P_{e_{jl}}|}{|P_{e_{ir}} \cup P_{e_{jl}}|}$$

For a pair of splits $(e_i, e_j)$, the score $s(e_i, e_j)$ is then defined by

$$s(e_i, e_j) = max\{min\{a_{ll}, a_{rr}\}, min\{a_{rl}, a_{lr}\}\} \tag{11}$$

Finally, the Nye Similarity score is the following:

$$\sum_{e_i \in \mathcal{T}_1} s(e_i f(e_i)) \tag{12}$$

where $e_i$ is an edge in $\mathcal{T}_1$, $f(e_i)$ is the corresponding edge of $e_i$ in $\mathcal{T}_2$, based on an alignment of edges between $\mathcal{T}_1$ and $\mathcal{T}_2$ that maximizes the quantity in Eq. (12).

Clustering Information Distance (CID) is another information-theoretic generalized RF distance metric. CID calculates a matching score for every split on the tree, which is based on Mutual Clustering Information[41]. Such metric, as previously compared[32], has the advantage of being continuous and relatively unbiased towards certain topologies. Using the three metrics, we are able to comprehensively evaluate the tree reconstruction methods, while RF distance provides a basic metric that is also widely used in other papers, the generalized metrics like Nye Similarity and CID provide more detailed and interpretable comparisons.

## Running LinRace and baseline algorithms

We have summarized the descriptions of the key packages in Table 1: lineage reconstruction methods (Startle, Cassiopeia, DCLEAR, LinTI-MaT), trajectory inference methods (Slingshot), data simulation (TedSim) and evaluation (TreeDist), that are used in this paper.

In addition, here are the major parameters and their settings for LinRace:

**Table 1 | A summary of key packages used in the paper**

| Package | Language | Version |
|---|---|---|
| Startle | Python | – |
| Cassiopeia | Python | 1.0.4 |
| DCLEAR | R | 1.0.10 |
| LinTIMaT | Java | – |
| Slingshot | R | 2.4.0 |
| TedSim | R | 0.0.0.9 |
| TreeDist | R | 2.4.1 |

- $\lambda_1$ and $\lambda_2$, set to be $\lambda_1 = 10$ and $\lambda_2 = 1$ in all tests.
- Number of genes kept after filtering: we keep the top 100 highly variable genes for simulated data.
- Maximum number of iterations for each local search is 500 in all results.
- Asymmetric division rate $p_a$ is set to 0.8 in all tests.

Running LinRace-IST also involves the steps of identifying cell states and inferring the cell state tree. $k$-means clustering method is used on the scRNA-seq data to identify cell states and Slingshot is applied to infer the cell state tree, based on the identified cell states. When using simulated data, the number of clusters for $k$-means is set to 7. The same number of clusters is used for the *C. elegans* data. For the scGESTALT dataset (ZF1-F3), the number of clusters for $k$-means is set to 20. For the mouse embryo data, we used annotated cluster IDs from the paper[5].

As the subtrees that LinRace local search aims to optimize can vary in size, we adopt a dynamic manner to set the number of iterations for each subtree. Denote the *n*th Catalan number[33] by $C(n)$, then the number of iterations used to optimize a subtree with *n* leaves is: $\min(C(n), max\_iter)$, where *max_iter* is the maximum number of iterations set as a parameter.

The parameter settings for baseline methods were mostly based on their default parameters. LinTIMaT has three major parameters: the number of genes *gc*, the number of mutation likelihood iterations *mi*, and the number of combined likelihood iterations *ci*. We set *gc* = 100 when running it on simulated data, the zebrafish dataset, and the mouse embryo dataset. For the *C. elegans* data, we used *gc* = 93 because the dataset has in total 93 genes. *mi* and *ci* were set to 20,000 for both simulated and real datasets.

For DCLEAR, we used the "kmer" mode and set the $k - mer$ length to 2 for all datasets, which is the default value for this parameter. We also used default values of other parameters for DCLEAR.

For Cassiopeia, Cassiopeia-greedy and Cassiopeia-hybrid are used for 1024 cells. For the Cassiopeia-hybrid, we set the convergence time limit for the ILP solver to be 1000, the maximum potential graph layer size to be 500 and set a cell cutoff between the top greedy solver and the bottom ILP solver to be 20. For the parameter *indel_priors* (which represents the probabilities of particular indels occurring), we have left it empty.

For Startle, we used the "NNI" mode and set the number of NNI iterations to 5000 for both TedSim datasets and C.elegans datasets. The main reason that we cannot further increase the number of iterations for TedSim datasets (1024 cells) is the running time of the algorithm. We used default values of other parameters for Startle-NNI.

More detailed descriptions of the parameter settings for running the simulation and the lineage reconstruction methods can be found in Supplementary Note 2.

### Analysis of lineage-expression relationships in early mouse embryo data
In order to explore cells' gene expressions under different subtrees, we perform Differential Expression(DE) analysis on the mouse embryo dataset. Based on the reconstructed lineage tree using LinRace (see Fig. 5), we cluster the cells by cutting the tree at depth 26, which results in a remaining tree (with cells at depth less than 26) and 12 subtrees as shown in Fig. 5b. We specifically look into the subtrees with relatively large populations of cells: subtree 9, 10, and 12. To investigate whether cells' lineage influences their gene expression profile, we looked into the Fore/Midbrain cell types in subtree 9, 10, and 12, and performed DE analysis. We used the "FindMarkers" function of Seurat, which adopts the Wilcoxon rank test to identify the DE genes between two groups of cells. The analysis is performed between the Fore/Midbrain cells under subtree 9, 10, and 12. As stated in the Results, no significant DE genes are found, indicating that lineage information is not reflected in the gene expression profiles under different clones.

Then, for the same Fore/Midbrain cells under subtree 9, 10, and 12, we compute their linear dimensionality reduction using PCA. Then, on the first 50 PCs, we computed pairwise Euclidean distances between pairs of single cells and grouped them based on their clone IDs. The results (Fig. 5e) show that the distances between the cells from different clones do not show significant differences from those of the same clone.

### Evaluating the quality of lineage barcode datasets
**Reconstruction potential.** Given a true lineage tree $\mathcal{T}^*$ and a character matrix $X$ (lineage barcodes) of both observed, leaf cells and hidden, ancestral cells (for TedSim datasets, the lineage barcodes of the ancestral nodes are known), we analyze the *reconstruction potential* which can be compared to the Robinson–Foulds distance of a lineage reconstruction method because both metrics are calculated based on the ratio of the splits on the lineage tree. The idea of the reconstruction potential is that for any given edge that splits all leaf cells into two sets, at least one mutation is required for this split to be reconstructed. Denote the true lineage tree as $\mathcal{T}^* = (V, E)$, where $V$ represents the cells on the lineage tree. The reconstruction potential $Q_r$ is given as follows:

$$Q_r(\mathcal{T}^*) = \frac{\sum_{(u,v) \in E} \mathbb{1}(X_u = X_v)}{||E||} \tag{13}$$

where $\mathbb{1}(x)$ is the characteristic function which returns 0 if $x$ is TRUE; 0 otherwise. $Q_r$ calculates the fraction of edges that contain at least a mutation.

**Homoplasy edge ratio.** Even if the edge contains a mutation, it is not guaranteed that the edge is recoverable due to barcode homoplasy. Given a true lineage tree $\mathcal{T}^*$ and a character matrix $X$ (lineage barcodes) of both observed, leaf cells and hidden, ancestral cells (for TedSim datasets, the lineage barcodes of the ancestral nodes are known), we quantify the barcode homoplasy effect in the data using the *homoplasy edge ratio*. For every edge being a homoplasy edge, we refer to the fact that all mutations induced on this edge are not unique. Then, the homoplasy edge ratio represents the ratio of homoplasy edges of all edges on the tree. Denote the true lineage tree as $\mathcal{T}^* = (V, E)$, where $V$ represents the cells on the lineage tree; and denote $X_b \backslash X_a = \{x \mid x \in X_b; x \notin X_a\}$ as the set of mutations that appear from the division event from cell $a$ to cell $b$. The homoplasy edge ratio $Q_h$ is given as follows:

$$Q_h(\mathcal{T}^*) = \frac{\sum_{(u,v) \in E} \mathbb{1}(X_v \backslash X_u \subset \bigcup_{(u',v') \in E \backslash (u,v)} X_{v'} \backslash X_{u'})}{||E||} \tag{14}$$

where $\mathbb{1}(x)$ is the characteristic function which returns 0 if $x$ is TRUE; 0 otherwise.

### Reporting summary
Further information on research design is available in the Nature Portfolio Reporting Summary linked to this article.

## Data availability

The *C. elegans* dataset from Liu et al. are available in the supplementary data from the original paper [https://doi.org/10.1016/j.cell.2009.08.044]. The mouse embryo dataset from Chan et al. was previously deposited in the Gene Expression Omnibus under accession number GSE117542. The ZF1-F3 dataset from scGESTALT was previously deposited in the Gene Expression Omnibus under accession number GSE105010. Large datasets (such as raw reconstructed trees for varying parameters) can be provided upon request. Source data are provided with this paper.

## Code availability

LinRace is available at https://github.com/ZhangLabGT/LinRace. The LinRace package is also citable with https://doi.org/10.5281/zenodo.10211293. TedSim is a simulator for paired lineage barcode and gene expression data available at https://github.com/Galaxeee/TedSim. Cassiopeia is an end-to-end pipeline for single-cell lineage tracing experiments available at https://github.com/YosefLab/Cassiopeia. DCLEAR is an R package for Distance-based Cell LinEAge Reconstruction (DCLEAR) available at https://github.com/ikwak2/DCLEAR. LinTI-MaT is a statistical method for reconstructing lineages from joint CRISPR-Cas9 mutations and single-cell transcriptomic data available at https://github.com/jessica1338/LinTIMaT. Startle is available at: https://github.com/raphael-group/startle.

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

## Acknowledgements

This work was partly supported by the US National Science Foundation DBI-2019771, DBI-2145736, and National Institutes of Health grant R35GM143070 (X.P., H.L., and X.Z.).

## Author contributions

X.P. and X.Z. conceived the study and designed the model. X.P. developed and implemented the algorithms. X.P., H.L., and P.P. performed the tests. X.P. and X.Z. wrote the manuscript and all authors approved it.

## Competing interests

The authors declare no competing interests.
