## [Peer Review File · Nature Communications]

LinRace: cell division history reconstruction of single cells using paired lineage barcode and gene expression dataREVIEWER COMMENTS

Reviewer #1 (Remarks to the Author):

Pan et al. presented as a new computational tool (LinRace) to reconstruct the developmental lineages based upon paired single cell RNA-seq and CRISPR / Cas9 based lineage recorders. LinRace first reconstructed a lineage tree from the unique barcodes (CRISPR / Cas9 scars) from the recorders, and used the scRNA-seq data from of cells with the identical barcode to derive the subtrees. They have evaluated the LinRace on synthetic data simulated by TedSim (also from the same group), and two real datasets on *C.elegans* development and the scGESTALT dataset. LinRace demonstrated better performance than state-of-the-art methods such as Cassiopeia and DCLEAR that reconstruct lineages solely from the CRISPR / Cas9 based lineage recorders, as well as LinTIMat that took both scRNA-seq and lineage recorders as the inputs. Overall, the manuscript is well written, and the analysis / benchmark is comprehensive. Here are a few issues the authors would consider to address:

1. The authors need to include more recently tools such as Startle (<https://www.biorxiv.org/content/10.1101/2022.12.18.520935v1.abstract>) into the benchmark.
2. Performance of tree reconstructions from Slingshot along should also be included. It is curious to see if the lineage tree reconstructed from scRNA-seq (by Slingshot) has comparable performance with Cassiopeia or DCLEAR. For example, a corresponding tree can be constructed from the cell-cell distance matrix using the latent representation.
3. Other tools that such as TreeVAE that also used both CRISPR / Cas9 recorders and scRNA-seq data also should be included in the benchmark.
4. The authors only examined the recorders with relatively small targets (16 and 9 targets in the synthetic data and *C. elegans* datasets, respectively). It is interesting to see the performance over more target sites. For example, 100 and 1,000 target sites were used in the DREAM challenge.
5. Section 4.7: Cassiopeia is written in Python.

Reviewer #2 (Remarks to the Author):

This articles addresses an interesting question of how to infer the lineage phylogeny with higher resolution. To do so, they developed a sophisticated framework to integrate the lineage and cell state information. In particularly, they considered the cell division history, the differentiation dynamics on the cell state manifold, and the consistency between these two processes. In doing so, the reconstructed lineage tree are shown to achieve higher accuracy than existing methods. Their idea is cool, and the paper is overall well-written. I have the following comments.

- 1, In order to correctly reconstruct the lineage phylogeny, it is important to consider barcode homoplasy, where two independent cells may spontaneously acquire the same mutations for lineage recording. In fact, target array editing with CARISPR-Cas9 generates outcomes (barcodes) with different frequencies,

with some being highly frequent, labeling many unrelated (non-clonal) cells, and others being rare and more specific to a particular clone. The issue of barcode homoplasmy becomes more severe when there is large scale dropouts of the target arrays, which is common to CARISPR-Cas9 based lineage recording. Indeed, in Supplementary Fig 1a, the largest clone (~1500 cells) has large deletions and only a single detected mutation. Assuming that all the cells sharing this barcode come from the same founder cell is likely wrong, because the only mutation in this barcode may be generated independently from many cells among these 'pseudo' clone. Besides, it is also unlikely that a single founder cell could expand to ~1500 cells, while most other clones only have a single cell. The authors should clearly discuss how to address this issue of barcode homoplasmy, due to having mutations with very different frequencies. Potentially, they can use their simulations to test this situation, and see how well their method performs in this context.

2, In order to simulate the cell division history and the differentiation dynamics together, the authors assume that the sampled leaf nodes can form a state manifold, and the ancestor cell states (from the internal nodes in the reconstructed lineage tree) can be found from the sampled cell states. In systems with asynchronous differentiation, like in blood, such an assumption could indeed hold. But for developmental systems with more synchronous differentiation, all cell states change rapidly over time, and the early ancestor cell states are no longer there at the later stage of development. This is an unstated limitation of this study. The authors should address this limitation. At least, they should explicitly state this limitation. In doing so, they may also need to temper their claim that they can infer the ancestor state, by adding the restrictive conditions.

3, There are two articles worth referencing in this paper.

3.1 Li et.al., bioRxiv 2023, titled "A mouse model with high clonal barcode diversity for joint lineage, transcriptomic, and epigenomic profiling in single cells". They explicitly addressed the issue of low barcode diversity due to large inter-site deletions in CARISPR-Cas9 editing by adding TdT to increase mutation diversity. This could help to reduce the issue of barcode homoplasmy in future applications.

3.2 Wang et.al., Nat. Biotech. 2023, titled "CoSpar identifies early cell fate biases from single-cell transcriptomic and lineage information". They similarly integrated lineage and state information. Instead of learning the lineage tree, they inferred the cell fate choice. In their study, they explicitly addressed how to cope with barcode homoplasmy in their algorithm.

Minor issues for Main text

Line 52, "Is it shown ..." → "It is shown.."

Line 113-116, the description of the clone size is very confusing here.

Line 369-374, how do you support your conjecture here about differentiation speed? I do not see any data.

Line 457, does "bifurcating tree" means that it will learn a topology of just two branches? Please clarify

Line 482 (Eq. 3), I believe the equation has typos. Also, $l(\cdot)$ function is not defined. Please clarify what this function is, and why it takes this form.

Line 495, what is x_i ? Gene expression vector of a single cell?

Line 561-569, I did not follow this part. Could you make it more clear?

Line 625, the definition of the $l()$ function should go earlier when it first appears.

Minor issues for Supplementary file

Supplementary Fig 2, clarify the figure legend: LinRace_0.1_1.

Could you provide a figure with varying cell number, and compare the performance of different methods at the same mutation rate, similar to Supplementary Figure 5?

Reviewer #3 (Remarks to the Author):

In “LinRace: cell division history reconstruction of single cells using paired lineage barcode and gene expression data”, the authors describe a new method to build cell lineage trees using CRISPR-Cas9 induced lineage barcode data supplemented with the simultaneously collected single cell RNA-seq data. The authors accurately describe the challenge in such data sets, specifically that the lineage barcodes alone often do not encode sufficient information to reconstruct the full lineage tree. To overcome this limitation, the authors leverage gene expression information to resolve the edges of the tree.

I believe this manuscript is well written and clearly presented. The new strategy LinRace serves a useful role in the field providing a route to include gene expression data with lineage barcode data only when lineage data is limited. In this way, the lineage data itself is prioritized in determining the cell lineage tree when it is available, which seems reasonable. The authors make the appropriate comparisons of their method to field standards (DCLEAR, Cassiopeia, and in some ways, the most directly comparable LinTIMat) using de novo simulated data sets and simulated data based on the known *C. elegans* lineage. Overall, their strategy performs the best using appropriate criteria. Moreover, the authors offer good insights into why methods generally perform better or worse over different mutation rates and why they believe their method outperforms LinTIMat. The data and methodology are sound.

Next, the authors apply their method to two real data sets where ground truth is not known, the zebrafish brain from scGESTALT and mouse embryos from the molecular recorder. In the former, they showcase an example where their strategy resolves lineages that undergo symmetric vs asymmetric cell division while in the latter, they describe the phenomenon of “partial consistency between transcriptome similarity and barcode similarity”. In both cases, the biological findings are limited but I do

not believe it detracts from this manuscript. Rather, it highlights how LinRace is useful in creating more detailed subtrees that may aid in future biological interpretation. For example, better tree topologies may lead to more accurate estimates of symmetric and asymmetric cell division likelihoods – a parameter that would affect the LinRace method itself.

Overall, I think this manuscript presents a reasonable method for utilizing gene expression data to complement lineage tracing data in a manner that is very useful for the field. I believe the description of methods and analysis performed in the manuscript is sufficient for replication.

Suggestions:

- Section 2.5 presents the lack of DE for the same cell state that exist in different lineages. Overall, this is not a surprising result and is in line with biological expectations. Nonetheless, their ability to detect DEs may simply be underpowered. I wonder if the intra-lineage gene expression distance for cells of the same state is lower than the inter-lineage distance across cells of the same state across the population of cells. I realize this suggestion can be a bit circular depending on the depth of tree examined since “neighbor distance likelihood” is used to establish the lineage. It might be interesting to look at regardless.

- I am curious if the authors can comment on how these following suggestions might change/improve performance (which could perhaps be added to discussion):

- o Alternate tree building method to make the tree backbone – perhaps DCLEAR or Cassiopeia.
- o Using time course scRNA-seq data instead of the scRNA-seq data collected with the lineage barcodes. Trajectories may be more accurately inferred when past cell states are more directly measured than relying on variability that exists within a time point. To calculate the GES, they use 3 terms, the first two of which utilizes only cell state information. They could use the cell state tree from scRNA-seq time course data sets, and still maintain the third term from the lineage traced data set.

Minor points:

Fig 1. It may be worthwhile to change the colors for cell states so that it is easier to distinguish between the black and navy, and red and pink

Line 176-8: I would recommend adding a half a sentence or a sentence about how to choose a root cell state outside of simulation environments.

Line 309: GESTAULT should be GESTALT (and also Fig 4 legend, line 591)

Fig 4a: Inner ring is barely visible

Line 334: This may be a lack of understanding on my part, but to my knowledge, terminal cells do not undergo mitosis. Is there a different word that can be used here to express cells that can only undergo symmetric divisions?

Responses to Reviewers' Comments

We thank the reviewers for their time to review our manuscript and their valuable comments.

Please find below the reviewers' comments (in black) and our pointwise responses (in blue). The changes in the manuscript are marked in red.

Reviewer #1

Pan et al. presented as a new computational tool (LinRace) to reconstruct the developmental lineages based upon paired single cell RNA-seq and CRISPR / Cas9 based lineage recorders. LinRace first reconstructed a lineage tree from the unique barcodes (CRISPR / Cas9 scores) from the recorders, and used the scRNA-seq data from of cells with the identical barcode to derive the subtrees. They have evaluated the LinRace on synthetic data simulated by TedSim (also from the same group), and two real datasets on C.elegans development and the scGESTAULT dataset. LinRace demonstrated better performance than state-of-the-art methods such as Cassiopeia and DCLEAR that reconstruct lineages solely from the CRISPR / Cas9 based lineage recorders, as well as LinTIMat that took both scRNA-seq and lineage recorders as the inputs. Overall, the manuscript is well-written, and the analysis/benchmark is comprehensive. Here are a few issues the authors would consider to address:

1. The authors need to include more recently tools such as Startle (<https://www.biorxiv.org/content/10.1101/2022.12.18.520935v1.abstract>) into the benchmark.

We thank the reviewer for suggesting that we include the recent method, Startle, in our comparison. Startle infers cell division trees from lineage barcode data by enforcing the "non-modifiability" property of CRISPR-Cas9 mutations.

Startle provides two modes, Startle-ILP and Startle-NNI. We encountered a running time issue with Startle-ILP (their preferred mode) on the scale of datasets we used: the solver could not find a valid solution with the default parameters. Based on communication with the authors of Startle, we used the Startle-NNI mode, which is much faster than Startle-ILP, although still slower than the other methods. As a result, we tested Startle-NNI on all simulated datasets except for those with 4096 cells.

We included the results of Startle-NNI in the comparisons of LinRace and baseline methods, using accuracy measurements RF distance, Nye similarity, and CID, in Figs. 2-3, and Supplementary Fig. 10 in the manuscript. The results show that LinRace outperforms Startle-NNI consistently.

In the manuscript, we have also added modifications and corresponding discussions to reflect that we included Startle as an additional baseline method. These changes are in Lines 68-69, 160, 221-225, 332, 364-366, 711-717, 765.

2. Performance of tree reconstructions from Slingshot along should also be included. It is curious to see if the lineage tree reconstructed from scRNA-seq (by Slingshot) has comparable performance with Cassiopedia or DCLEAR. For example, a corresponding tree can be constructed from the cell-cell distance matrix using the latent representation.

We thank the reviewer for providing two suggestions on using only gene expression data to reconstruct the cell lineage tree.

- (1) Although Slingshot is widely used for the task “trajectory inference” which is to infer trajectory between cell states (and thus is used in LinRace to infer the *cell state tree*), it is not suited to infer cell division trees, which is the final goal of LinRace. We clarified the relationships between the cell state tree used in LinRace and the cell division tree in the manuscript (e.g. Lines 96-100).

We understand that this can be confusing as the term “lineage tree” is sometimes used to refer to the trajectory between cell states, while in the context of cell division events, a lineage means the tree representing the cell division history. To avoid this confusion, we clarified the definition of “cell lineage tree” and trajectory inference in the paper (Lines 39-42, 117-121).

The main reason that Slingshot can not be used to infer a cell division tree is the following: A common application of Slingshot is that given k clusters of cells, Slingshot infers a tree with k nodes. Similarly, if one cell is treated as a cluster, given the input of n cells, Slingshot infers a tree with n nodes, where some are internal nodes and some are leaf nodes of the tree. When inferring a cell division tree, all n cells need to be leaf nodes. In other words, Slingshot estimates *pseudotime* while the task of inferring cell division trees considers the time from the root to leaves as real-time.

- (2) Based on the suggestion of using the cell-cell distance matrix obtained from cell latent representations to reconstruct the cell division tree, we implemented and tested this approach (called NJ-exp) on the TedSim simulated datasets (1024 cells and 16 target sites). Specifically, we first performed PCA on the scRNA-seq count matrix, then used the first 50 PCs to calculate pairwise Euclidean distance between single cells, and reconstructed a NJ tree from the pairwise distance matrix.

The results in terms of all three metrics, RF distance, Nye similarity, and Clustering Info Distance are shown below (orange boxes represent the expression-only approach NJ-exp). We can observe that the accuracy of NJ-exp is clearly worse than other methods that use lineage barcode information. In particular, the RF distance is close to 1

(with a very small standard deviation). This confirms that gene expression data alone is not sufficient to reconstruct cell division histories. We reckon that the reasons for this are both technical and biological: technically the scRNA-seq data measurements are highly noisy, and biologically the transcriptomes of cells do not necessarily reflect the whole cell division lineages.

Figure R1. Performance comparison between the gene-expression-only method “NJ-exp” and other methods that use barcode information.

3. Other tools such as TreeVAE that also used both CRISPR / Cas9 recorders and scRNA-seq data should also be included in the benchmark.

We thank the reviewer for this comment. Although TreeVAE (<https://www.biorxiv.org/content/10.1101/2021.05.28.446021v1.abstract>) also takes the lineage barcode and gene expression data of single cells as input, which is the same as the input the LinRace, the goal and output of TreeVAE is different from LinRace and other baseline methods included in our current comparison. TreeVAE employs existing methods (e.g. Cassiopeia) to reconstruct the cell division tree and then focuses on the inference of the gene

expression profile of ancestral cells. Cassiopeia, the tree reconstruction method used in TreeVAE, has been included in our benchmark. Although TreeVAE can not be compared with LinRace and other baseline methods included in our benchmark, we have cited TreeVAE as a method that takes both lineage barcode and gene expression data of single cells as input (Lines 83-84 in the manuscript).

4. The authors only examined the recorders with relatively small targets (16 and 9 targets in the synthetic data and *C. elegans* datasets, respectively). It is interesting to see the performance over more target sites. For example, 100 and 1,000 target sites were used in the DREAM challenge.

We thank the reviewer for this comment. First, the reason that we used a relatively small number of targets (16 for 1024 cells, 64 for 4096 cells) to test the methods is that in real datasets, the number of targets is small. Therefore, the results on simulated data can better resemble the case on real data. For example, scGESTALT (Raj et al. 2018) has 9 target sites, the mouse embryo dataset in (Chan et al. 2019) has 18 target sites, and the data in (Quinn et al. 2021) has 10 to 30 target sites.

That being said, we agree that it is useful to demonstrate the performances of the methods on datasets with a larger number of target sites. We performed comparisons of LinRace and baseline methods (LinRace-IST, Cassiopeia-hybrid, DCLEAR-kmer, LinTIMaT) using 128 targets and 1024 cells, and included the results in Supplementary Fig. 4. Startle was not able to complete certain runs (after 12 hours), so results for Startle in these runs were not included. On barcode data with 128 target sites, LinRace still outperforms other methods, and overall, the tree accuracy obtained with 128 target sites is slightly higher than that obtained with 16 target sites.

We have added discussions in the manuscript to reflect the new results on barcode data with 128 target sites (Lines 212, 225-228).

5. Section 4.7: Cassiopedia is written in Python.

Thanks for the correction, and we have corrected this in the manuscript (Line 678).

Reviewer #2

This article addresses an interesting question of how to infer the lineage phylogeny with higher resolution. To do so, they developed a sophisticated framework to integrate the lineage and cell state information. In particular, they considered the cell division history, the differentiation dynamics on the cell state manifold, and the consistency between these two processes. In doing so, the reconstructed lineage tree are shown to achieve higher accuracy than existing methods. Their idea is cool, and the paper is overall well-written. I have the following comments.

1, In order to correctly reconstruct the lineage phylogeny, it is important to consider barcode homoplasy, where two independent cells may spontaneously acquire the same mutations for lineage recording. In fact, target array editing with CARISPR-Cas9 generates outcomes (barcodes) with different frequencies, with some being highly frequent, labeling many unrelated (non-clonal) cells, and others being rare and more specific to a particular clone. The issue of barcode homoplasy becomes more severe when there is large scale dropouts of the target arrays, which is common to CARISPR-Cas9 based lineage recording. Indeed, in Supplementary Fig 1a, the largest clone (~1500 cells) has large deletions and only a single detected mutation. Assuming that all the cells sharing this barcode come from the same founder cell is likely wrong, because the only mutation in this barcode may be generated independently from many cells among these 'pseudo' clone. Besides, it is also unlikely that a single founder cell could expand to ~1500 cells, while most other clones only have a single cell. The authors should clearly discuss how to address this issue of barcode homoplasy, due to having mutations with very different frequencies. Potentially, they can use their simulations to test this situation, and see how well their method performs in this context.

This is a great point. To analyze barcode homoplasy in the data we used in our tests and show how LinRace contributes to alleviating the homoplasy issue, we added Supplementary Fig. 8 to the manuscript. Our response to this comment includes the following points:

1. Although we did not explicitly discuss barcode homoplasy in our original submission, the simulated data we used in the tests models the exact two factors mentioned in the reviewer's comments that lead to barcode homoplasy: (1) the biased frequency of mutations in the barcode, and (2) the dropout events. (Details described in Methods "Simulating synthetic datasets with TedSim", Lines 610-613, 618-631.) By using simulated data modeling these factors, the accuracy of tree reconstruction methods reflects the performance of these methods on real data with large pseudo clones.
2. In other words, the results we reported in the manuscript are indeed on datasets with the popular existence of barcode homoplasy, and we showed that LinRace performs better than other methods on such datasets. Supplementary Fig. 1b shows the large pseudo-clones in our simulated data. Next, we use a different measurement, "homoplasy edge ratio", to show the high frequency of the same mutation occurring on multiple lineages during the cell division process. Since with simulation we can obtain the true cell division tree and the barcode mutation on each edge of the tree, we can calculate the "homoplasy edge ratio" in the data as follows: for each edge connecting a parent cell and a daughter cell, we call it a "homoplasy edge" if there is no unique mutation on this edge (a "unique mutation" means the same mutation at the same target site does not happen at any other edge on the tree). The homoplasy edge ratio is calculated as the number of homoplasy edges divided by the total number of edges in the tree (Supplementary Fig. 8a).

From Supplementary Fig. 8a, we can see that the homoplasy ratio in data with dropouts

is much larger than in data without dropouts. Also, when there are dropouts, the trend of homoplasmy edge ratio changes over the mutation rate is consistent with that of the accuracy of lineage reconstruction methods shown in Fig. 2a-d. This indicates that the homoplasmy ratio we calculate can correlate with barcode data quality. With the existence of dropouts, the homoplasmy ratio can be as high as 0.75.

3. The results we included in the manuscript (Fig. 2a-d, Fig. 3, Supplementary Fig. 4) confirm that LinRace has superior performance compared to baseline methods on datasets with high amounts of barcode homoplasmy. Here we show examples of subtrees where LinRace helps to correct homoplasmy to a certain extent, thanks to the use of gene expression data. From one of our simulated datasets used in Fig. 2, we take a pseudo-clone with 13 cells. These cells have the same barcode but originate from different clones in the true tree (Supplementary Fig. 8b). When considering only these 13 cells, we can obtain the true tree among them (Supplementary Fig. 8c). Since they have the same barcode, lineage-barcode-based methods such as NJ, DCLEAR, and Cassiopeia can only randomly guess the tree of these cells (an example of the random tree is in Supplementary Fig. 8e). The LinRace reconstructed tree of these cells is shown in Supplementary Fig. 8d. Visually, we see that the LinRace reconstructed tree is able to recover correct branches such as cell 3 and cell 4, cell 13 and cell 14. We also show that the LinRace reconstructed tree of these cells is more accurate than the true tree (Supplementary Fig. 8f) in terms of the CID metric. When using RF distance, the random trees mostly have a RF distance of 1 while the LinRace reconstructed tree has a RF distance of 0.8. We also use heatmaps to provide cell-cell distance on the tree for the true tree, LinRace inferred tree, and a random tree (Supplementary Fig. 8f) to show that the LinRace inferred tree is closer to the true tree compared to a random tree.

Overall, barcode homoplasmy presents a significant challenge for methods of reconstructing cell division trees. The use of gene expression data in LinRace helps to partially recover the relative positions of cells with the same barcode in the tree. However, since the gene expression of cells is dominated by cell types rather than lineages, there is a limit to how much gene expression data can help. Moreover, for computational efficiency, LinRace runs tree refinement on local subtrees. To fully recover the positions of cells in the global tree, optimization on the global tree structure is needed, but methods that perform whole-tree structure optimization did not prove successful (e.g. LinTIMaT and Startle) due to the vast search space of the whole tree.

In the manuscript, we have added this discussion in Lines 278-313. We added the details of calculating the homoplasmy edge ratio to the Methods section (Lines 745-754).

2, In order to simulate the cell division history and the differentiation dynamics together, the authors assume that the sampled leaf nodes can form a state manifold, and the ancestor cell states (from the internal nodes in the reconstructed lineage tree) can be found from the sampled cell states. In systems with asynchronous differentiation, like in blood, such an assumption could indeed hold. But for developmental systems with more synchronous differentiation, all cell states change rapidly over time, and the early ancestor cell states are no

longer there at the later stage of development. This is an unstated limitation of this study. The authors should address this limitation. At least, they should explicitly state this limitation. In doing so, they may also need to temper their claim that they can infer the ancestor state, by adding the restrictive conditions.

This is a great point. Indeed, for developmental systems where all cell states change rapidly over time, the early ancestor cell states are unlikely to exist in the scRNA-seq data, therefore, these states can not be observed in the data, nor can they be reconstructed by trajectory inference methods.

If the scRNA-seq dataset does not cover such cell states, then LinRace can not incorporate these cell states, as LinRace uses the cell state tree inferred from data. However, if a prior cell state tree that includes these cell states is given, LinRace can assign cell states for ancestral cells as the cell states missing from scRNA-seq data.

We have added a discussion about the limitation of utilizing the LinRace method for different systems of biology (main manuscript Line 471-480). Also, we acknowledge the restrictive conditions when stating that LinRace infers cell states for ancestral cells (Lines 145, 380-381).

3, There are two articles worth referencing in this paper.

3. 1 Li et.al., bioRxiv 2023, titled “A mouse model with high clonal barcode diversity for joint lineage, transcriptomic, and epigenomic profiling in single cells”. They explicitly addressed the issue of low barcode diversity due to large inter-site deletions in CARISRP-Cas9 editing by adding TdT to increase mutation diversity. This could help to reduce the issue of barcode homoplasmy in future applications.

3.2 Wang et.al., Nat. Biotech. 2023, titled “CoSpar identifies early cell fate biases from single-cell transcriptomic and lineage information”. They similarly integrated lineage and state information. Instead of learning the lineage tree, they inferred the cell fate choice. In their study, they explicitly addressed how to cope with barcode homoplasmy in their algorithm.

We thank the reviewer for suggesting these relevant papers. We have cited these two papers in our revised manuscript (Lines 34, 482 for Li *et al.*, and Line 83-84 for Wang *et al.*).

We did not discuss Wang *et al.* specifically in terms of their ability to cope with barcode homoplasmy, as the method CoSpar does not aim to output the cell division tree, thus barcode homoplasmy affects their task differently from how barcode homoplasmy affects cell division tree reconstruction methods. Although in their results they showed the robustness of their method to homoplasmy, their method did not appear to directly target the homoplasmy issue.

Minor issues for Main text

Line 52, “Is it shown ...” → “It is shown..”

This has been fixed (now Line 55).

Line 113-116, the description of the clone size is very confusing here.

We thank the reviewer for pointing this out. This description aims to emphasize that there are a large number of cells that share the same barcode. Therefore, the barcode information alone can not distinguish these cells in the cell division tree. We rewrote this part (Lines 126-132) and also added new figures to help demonstrate the large clone size in real data (Supplementary Figure 1a).

Line 369-374, how do you support your conjecture here about differentiation speed? I do not see any data.

We thank the reviewer for raising this question. This paragraph on differentiation speed was intended to be an extended discussion in addition to all other results shown in the manuscript. It is related to the Reviewer's Comment 2, that certain biological systems go through asynchronous differentiation, and cells at different states exist in the scRNA-seq data. In the manuscript, we conjectured that cells at a later stage of development originate from lineages with fast differentiation speed. We realized that it is more appropriate to rephrase the conjecture as "cells at a later stage of development originate from lineages with more cell state transitions", as "differentiation speed" can be cell-specific and requires additional data to reach a conclusion, and the Reviewer is right that we did not show any data on differentiation speed.

In the revised manuscript, we removed this paragraph of extended discussion, as it is not part of our major results and is not rigorously supported with results.

Line 457, does "bifurcating tree" means that it will learn a topology of just two branches? Please clarify

Yes. To make this more clear, we have (1) changed all "bifurcating tree" into "binary tree" as the latter may be familiar to a larger audience; (2) added an explanation of this term "(which means every non-leaf node has exactly two children nodes)" when it first appears in the "Methods" section (Line 540). Also, we have the same explanation of this term in Section 2.1 (Line 150).

Line 482 (Eq. 3), I believe the equation has typos. Also, $I(\cdot)$ function is not defined. Please clarify what this function is, and why it takes this form.

We thank the reviewer for pointing this out. We have corrected Eq. 3 and also added other equations and descriptions to improve the clarity of this part of the calculation (Lines 551-585). The $I(x)$ function is the characteristic function that returns 1 if x is TRUE; 0 otherwise. The main idea of the calculation of state transition likelihood is as follows: intuitively, the likelihood of transition from State S_i to State S_j is high if the distance from S_i to S_j on the cell state tree is small, and low if the distance from S_i to S_j on the cell state tree is large. Since our procedure of ancestral state inference (described in Lines 527-537) implicitly ensures that in the candidate cell division tree, the edges with state transition between states with shorter distances in the cell state tree are more frequent than edges with state transition between states with longer

distance in the cell state tree, we use the frequency of edges in the candidate cell division tree with the same state distance as that from S_i to S_j as the likelihood of state transition from S_i to S_j .

Line 495, what is x_i ? Gene expression vector of a single cell?

Yes, x_i is the gene expression vector of cell i . We clarified the notations in the manuscript (Lines 557-558).

Line 561-569, I did not follow this part. Could you make it more clear?

We realized that this part of the description of the Nye similarity lacks clarity. We have rewritten the derivations of Nye similarity and added context and description on notations to improve the readability (Lines 581-582).

Line 625, the definition of the $l()$ function should go earlier when it first appears.

This is a great point. We have added the definition of the characteristic function when it first appears (Line 565).

Minor issues for Supplementary file

Supplementary Fig 2, clarify the figure legend: LinRace_0.1_1.

We now have updated the Figure Description under Supplementary Fig 2 to clarify this point: the first number in the legend is the weight for asymmetric division likelihood, and the second number is the weight for neighbor distance likelihood.

Could you provide a figure with varying cell number, and compare the performance of different methods at the same mutation rate, similar to Supplementary Figure 5?

This is a good point. We now have included a figure benchmarking performances of selected methods with varying numbers of cells in Supplementary Figure 6. We use the mutation rate = 0.1. As expected, the performance of all methods decreases with the increase in the number of cells, while LinRace outperforms all other methods. In the manuscript, we added discussions on this additional result in Lines 236-239.

Reviewer #3

In "LinRace: cell division history reconstruction of single cells using paired lineage barcode and gene expression data", the authors describe a new method to build cell lineage trees using CRISPR-Cas9 induced lineage barcode data supplemented with the simultaneously collected single cell RNA-seq data. The authors accurately describe the challenge in such data sets,

specifically that the lineage barcodes alone often do not encode sufficient information to reconstruct the full lineage tree. To overcome this limitation, the authors leverage gene expression information to resolve the edges of the tree.

I believe this manuscript is well written and clearly presented. The new strategy LinRace serves a useful role in the field providing a route to include gene expression data with lineage barcode data only when lineage data is limited. In this way, the lineage data itself is prioritized in determining the cell lineage tree when it is available, which seems reasonable. The authors make the appropriate comparisons of their method to field standards (DCLEAR, Cassiopeia, and in some ways, the most directly comparable LinTIMat) using de novo simulated data sets and simulated data based on the known *C. elegans* lineage. Overall, their strategy performs the best using appropriate criteria. Moreover, the authors offer good insights into why methods generally perform better or worse over different mutation rates and why they believe their method outperforms LinTIMat. The data and methodology are sound.

Next, the authors apply their method to two real data sets where ground truth is not known, the zebrafish brain from scGESTALT and mouse embryos from the molecular recorder. In the former, they showcase an example where their strategy resolves lineages that undergo symmetric vs asymmetric cell division while in the latter, they describe the phenomenon of “partial consistency between transcriptome similarity and barcode similarity”. In both cases, the biological findings are limited but I do not believe it detracts from this manuscript. Rather, it highlights how LinRace is useful in creating more detailed subtrees that may aid in future biological interpretation. For example, better tree topologies may lead to more accurate estimates of symmetric and asymmetric cell division likelihoods – a parameter that would affect the LinRace method itself.

Overall, I think this manuscript presents a reasonable method for utilizing gene expression data to complement lineage tracing data in a manner that is very useful for the field. I believe the description of methods and analysis performed in the manuscript is sufficient for replication.

Suggestions:

- Section 2.5 presents the lack of DE for the same cell state that exist in different lineages. Overall, this is not a surprising result and is in line with biological expectations. Nonetheless, their ability to detect DEs may simply be underpowered. I wonder if the intra-lineage gene expression distance for cells of the same state is lower than the inter-lineage distance across cells of the same state across the population of cells. I realize this suggestion can be a bit circular depending on the depth of tree examined since “neighbor distance likelihood” is used to establish the lineage. It might be interesting to look at regardless.

This is a great suggestion to complement the DE results. As suggested, we focused on cells in the Fore/MidBrain cell type, and compared intra-clone gene expression distances with inter-lineage distances. To do this, we took 3 clones from the Fore/MidBrain cell cells (clones 9,

10, and 12 in Fig. 5d), and calculated for the following pairs of inter-clone or intra-clone distances: Clone 9 vs Clone 9 (“9_9” in the figure), Clone 9 vs Clone 10 (9_10), Clone 9 vs Clone 12 (9_12), Clone 10 vs Clone 10 (10_10), Clone 10 vs Clone 12 (10_12), Clone 12 vs Clone 12 (12_12). Each pair of comparisons corresponds to a box in the boxplot, which is included in the manuscript as Fig. 5e.

First, we performed PCA on the gene expression data and kept the top 50 PCs for dimensionality reduction. Then to compare Clone i with Clone j , we calculated all pairwise Euclidean distances between any cell in Clone i and any cell in Clone j , and used these distances to generate the corresponding boxplot.

As shown in Figure 5e, there are no observable differences between the intra-clone distances and the inter-clone distances. The use of the “neighbor distance likelihood” mentioned in the reviewer’s comment can only contribute to the differences between intra-clone distances and the inter-clone distances. The fact that no clear differences are observed indicates that other factors, like the barcode similarity and state transition likelihood, are more dominant factors than the “neighbor distance likelihood” in forming the reconstructed tree. This additional result on gene expression distances is in line with the hypothesis in our manuscript and other publications (Packer et al. 2019) that cells’ gene expression signatures are dominated by the cell types instead of lineages.

We have added corresponding discussions on this result in the manuscript (Lines 440-443).

- I am curious if the authors can comment on how these following suggestions might change/improve performance (which could perhaps be added to discussion):

- o Alternate tree building method to make the tree backbone – perhaps DCLEAR or Cassiopeia.
- o Using time course scRNA-seq data instead of the scRNA-seq data collected with the lineage barcodes. Trajectories may be more accurately inferred when past cell states are more directly measured than relying on variability that exists within a time point. To calculate the GES, they use 3 terms, the first two of which utilizes only cell state information. They could use the cell state tree from scRNA-seq time course data sets, and still maintain the third term from the lineage traced data set.

We thank the reviewer for these suggestions. Please see the following responses to the two suggestions separately:

1. To find out how the methods used to build the backbone tree affect the final performance of LinRace, we conducted tests to use DCLEAR (using mode DCLEAR-kmer) and Cassiopeia (using mode Cassiopeia-greedy) to build the backbone tree (Fig. 2e, Supplementary Fig. 5). The results show that LinRace can improve upon the backbone tree inferred by DCLEAR-kmer and Cassiopeia-greedy. While using different backbone methods among these three choices (DCLEAR-kmer, Cassiopeia-greedy, and NJ) yields comparable results, the default combination we use (NJ and LinRace)

performs slightly better than other choices. We have added discussions on these results in the manuscript (Lines 230-236).

2. We agree that time-course scRNA-seq data helps to reveal past cell states that may not be captured in one scRNA-seq sample at one time point. Furthermore, we have added discussions in the manuscript to acknowledge such scenarios and the limitations of applying LinRace to datasets where the scRNA-seq data does not capture all or most of the cell states in the developmental process (Lines 471-475, 145, 380-381).

Time course scRNA-seq data will allow us to obtain a more complete set of cell states, with partial information on the temporal order of these cell states, which allows for the reconstruction of a more precise and comprehensive cell state tree. In this case, as suggested by the reviewer, only the cell state tree needs to be replaced by the one learned from the time course data, and other parts of LinRace can remain the same. We have added this discussion in the manuscript (Lines 477-479).

Minor points:

Fig 1. It may be worthwhile to change the colors for cell states so that it is easier to distinguish between the black and navy, and red and pink

Thanks for this comment. For Fig. 1, we used colors from the default rainbow color palette from base R. We have tried different options and the seven colors we used from the rainbow palette are the most distinguishable (See Figure R2). The candidate color options are as follows, and the leftmost one is the panel we use in Fig. 1.

We did modify Fig. 1 to make it easier to distinguish the colors, which is that we changed the border colors of the squares (and circles) to be the same as their respective fill colors. After removing the black border colors, we think it is now easier to distinguish black from navy, and red from pink.

Figure R2. Colour palettes visualization for cell types and lineage barcode mutation annotations in Fig. 1

Line 176-8: I would recommend adding a half a sentence or a sentence about how to choose a root cell state outside of simulation environments.

We have added one sentence on the common practice of trajectory inference on real data (Lines 195-196). In the case of Slingshot, a root cell state needs to be provided to the algorithm based on domain knowledge.

Line 309: GESTAULT should be GESTALT (and also Fig 4 legend, line 591)

Thanks for the correction. We have modified the text in our manuscript accordingly (now Lines 386, 678, Fig. 4 description).

Fig 4a: Inner ring is barely visible

Since the inner ring provides similar information as the color of the outer ring, and the only difference is that the inner ring shows sub-cell types, we have removed the inner ring for clearer visualization in both Fig 4a and Supplementary Fig. 12, and updated the figure descriptions.

Line 334: This may be a lack of understanding on my part, but to my knowledge, terminal cells do not undergo mitosis. Is there a different word that can be used here to express cells that can only undergo symmetric divisions?

We thank the reviewer for pointing this out. We have changed the use of “terminal cell type” to the specific cell type we observed in the figures (the Forebrain or Midbrain cell type) to avoid making general statements that are not well-supported (Lines 405, 411,412 in the manuscript).

References

- Chan, Michelle M., Zachary D. Smith, Stefanie Grosswendt, Helene Kretzmer, Thomas M. Norman, Britt Adamson, Marco Jost, Jeffrey J. Quinn, Dian Yang, Matthew G. Jones, Alex Khodaverdian, Nir Yosef, Alexander Meissner, and Jonathan S. Weissman. 2019. “Molecular Recording of Mammalian Embryogenesis.” *Nature*, May. doi:10.1038/s41586-019-1184-5.
- Packer, Jonathan S., Qin Zhu, Chau Huynh, Priya Sivaramakrishnan, Elicia Preston, Hannah Dueck, Derek Stefanik, Kai Tan, Cole Trapnell, Junhyong Kim, Robert H. Waterston, and John I. Murray. 2019. “A Lineage-Resolved Molecular Atlas of *C. Elegans* Embryogenesis at Single-Cell Resolution.” *Science* 365 (6459). science.sciencemag.org. doi:10.1126/science.aax1971.
- Quinn, Jeffrey J., Matthew G. Jones, Ross A. Okimoto, Shigeki Nanjo, Michelle M. Chan, Nir Yosef, Trevor G. Bivona, and Jonathan S. Weissman. 2021. “Single-Cell Lineages Reveal the Rates, Routes, and Drivers of Metastasis in Cancer Xenografts.” *Science* 371 (6532). doi:10.1126/science.abc1944.
- Raj, Bushra, Daniel E. Wagner, Aaron McKenna, Shristi Pandey, Allon M. Klein, Jay Shendure, James A. Gagnon, and Alexander F. Schier. 2018. “Simultaneous Single-Cell Profiling of Lineages and Cell Types in the Vertebrate Brain.” *Nature Biotechnology* 36 (5): 442–50. doi:10.1038/nbt.4103.

REVIEWERS' COMMENTS

Reviewer #1 (Remarks to the Author):

The authors have thoroughly responded to all my concerns, incorporating additional tools and datasets in their benchmark. I believe the manuscript is now ready for publication.

Reviewer #2 (Remarks to the Author):

The authors have addressed my concerns!

Reviewer #3 (Remarks to the Author):

The authors have addressed my concerns and have updated their manuscript accordingly. I am satisfied with their revision.